# LogicXGNN: Grounded Logical Rules for Explaining Graph Neural Networks

**Chuqin Geng**
Department of Computer Science
McGill University; University of Toronto
`chuqin.geng@mail.mcgill.ca; chuqin.geng@mail.utoronto.ca`

**Ziyu Zhao**          **Zhaoyue Wang**          **Haolin Ye**          **Yuhe Jiang**
McGill University      McGill University         McGill University      University of Toronto

**Xujie Si**
Department of Computer Science
University of Toronto
`six@cs.toronto.edu`

## Abstract

Existing rule-based explanations for Graph Neural Networks (GNNs) provide global interpretability but often optimize and assess fidelity in an intermediate, uninterpretable concept space, overlooking grounding quality for end users in the final subgraph explanations. This gap yields explanations that may appear faithful yet be unreliable in practice. To this end, we propose LogicXGNN, a post-hoc framework that constructs logical rules over reliable predicates explicitly designed to capture the GNN's message-passing structure, thereby ensuring effective grounding. We further introduce data-grounded fidelity ($Fid_{\mathcal{D}}$), a realistic metric that evaluates explanations in their final-graph form, along with complementary utility metrics such as coverage and validity. Across extensive experiments, LogicXGNN improves $Fid_{\mathcal{D}}$ by over 20% on average relative to state-of-the-art methods while being 10–100× faster. With strong scalability and utility performance, LogicXGNN produces explanations that are faithful to the model's logic and reliably grounded in observable data. Our code is available at `https://github.com/allengeng123/LogicXGNN`.

## 1 Introduction

Graph Neural Networks (GNNs) have emerged as powerful tools for modeling and analyzing graph-structured data, achieving remarkable performance across diverse domains, including drug discovery (Xiong et al., 2021; Liu et al., 2022; Sun et al., 2020), fraud detection (Rao et al., 2021), and recommender systems (Chen et al., 2022). Despite their success, GNNs share the black-box nature inherent to neural networks, posing challenges to deployment in high-reliability applications such as healthcare (Amann et al., 2020; Bussmann et al., 2021).

To address this, numerous explanation methods have been developed to uncover the decision-making mechanisms of GNNs. However, most existing approaches are limited to providing local explanations tailored to specific input instances or rely on input-feature attributions for interpretability (Pope et al., 2019; Ying et al., 2019; Vu & Thai, 2020; Lucic et al., 2022; Tan et al., 2022). A complementary line of work focuses on global explanations that characterize overall model behavior using rule-based approaches (Xuanyuan et al., 2023; Azzolin et al., 2023; Armgaan et al., 2024). These methods map relevant substructures into an *intermediate, abstract* concept space and then optimize logical formulas over these concepts to produce class-discriminative explanations. For interpretability, the abstract concepts are subsequently grounded in representative subgraphs, which serve as the final explanations presented to end users.

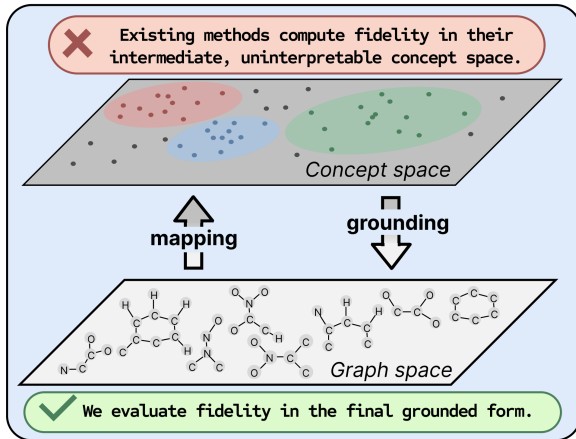
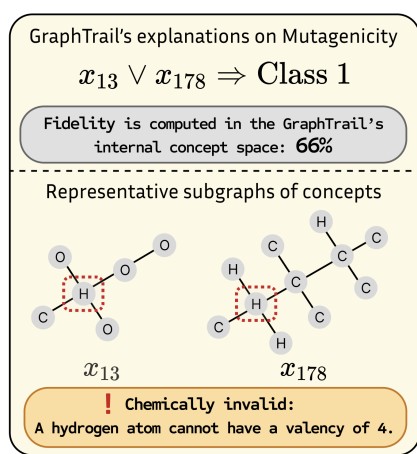

(a) Existing methods compute fidelity in the concept space.

(b) High fidelity yet meaningless explanations.

Figure 1: Existing methods such as GRAPHTRAIL compute fidelity in an uninterpretable concept space while overlooking the grounding quality of final subgraph explanations presented to end users.

While intuitive, this grounding step can introduce *unfaithfulness* and *unreliability*: methods may (i) reconstruct invalid subgraphs by mismatching node attributes and concept structure, or (ii) select plausible yet poorly representative subgraphs as post hoc rationalizations. This echoes a critical concern in explainable AI (Laugel et al., 2019; Lipton, 2018), where explanations may reflect learned artifacts rather than genuine data evidence. More importantly, these approaches optimize and evaluate fidelity—the degree to which explanations align with model predictions—*in their intermediate concept space*, neglecting issues with the ill-grounded subgraph explanations ultimately presented, as illustrated in Figure 1(a). This oversight risks producing explanations that appear highly faithful yet fail to reflect concrete, observable patterns in the data, thereby undermining both usability and trustworthiness for end users (Camburu et al., 2020). For example, although the state-of-the-art method GRAPHTRAIL (Armgaan et al., 2024) reports a $66\%$ fidelity score in its concept space on the Mutagenicity dataset (Debnath et al., 1991), its final explanations are *entirely ungrounded*: not a single explanation subgraph is chemically valid, and none matches an instance in the dataset (see Figure 1(b)). To address this gap, we propose a framework for evaluating rule-based explanations in their final, grounded form. Our approach centers on data-grounded fidelity ($Fid_{\mathcal{D}}$), a metric that assesses fidelity directly on the final subgraph explanations, supplemented by utility metrics such as coverage and validity. Under these criteria, the performance of existing methods drops dramatically, highlighting the need for explanations that are both faithful and truly interpretable.

To this end, we propose LOGICXGNN, a novel post hoc framework for constructing explanation rules over reliable predicates. These predicates are explicitly designed to capture the structural patterns induced by the GNN's message-passing mechanism, providing a solid foundation for reliable grounding. As a result, LOGICXGNN not only generates a rich set of representative subgraphs but also learns generalizable grounding rules for each predicate, addressing unreliable grounding in existing methods. Furthermore, our data-driven approach is highly efficient and demonstrates superior scalability on large real-world datasets, advantages we validate through extensive experiments. In summary, our key contributions are:

- We identify a key issue in existing rule-based explanation methods for GNNs: they optimize and evaluate fidelity in an intermediate, uninterpretable concept space without proper data grounding, which undermines usability and trustworthiness. To quantify this effect, we introduce $Fid_{\mathcal{D}}$, computed directly on the final-graph explanations presented to end users.

- We introduce LOGICXGNN, a novel framework for generating faithful and interpretable logical rule-based explanations for GNNs. Unlike existing methods, LOGICXGNN preserves structural patterns from message passing, enabling effective grounding that produces not only more representative subgraphs but also generalizable grounding rules.

- Our experimental results show that LOGICXGNN significantly outperforms existing methods, achieving an average improvement of over 20% in *Fid$_\mathcal{D}$* while being *10–100×* faster in runtime. Additional metrics, including *coverage*, *stability*, and *validity*, further confirm the superior practical utility of our generated explanations over existing methods.

## 2   PRELIMINARY

### 2.1   GRAPH NEURAL NETWORKS FOR GRAPH CLASSIFICATION

Consider a graph $G = (V_G, E_G)$, where $V_G$ represents the set of nodes and $E_G$ represents the set of edges. For the graph dataset $\mathcal{G}$, let $\mathcal{V}$ and $\mathcal{E}$ denote the sets of vertices and edges across all graphs in $\mathcal{G}$, respectively, with $|\mathcal{V}| = n$. Each node is associated with a $d_0$-dimensional feature vector, and the input features for all nodes are represented by a matrix $\mathbf{X} \in \mathbb{R}^{n \times d_0}$. An adjacency matrix $\mathbf{A} \in \{0,1\}^{n \times n}$ is defined such that $\mathbf{A}_{ij} = 1$ if an edge $(i,j) \in \mathcal{E}$ exists, and $\mathbf{A}_{ij} = 0$ otherwise. A graph neural network (GNN) model $M$ learns to embed each node $v \in \mathcal{V}$ into a low-dimensional space $\mathbf{h}_v \in \mathbb{R}^{d_L}$ through an iterative message-passing mechanism over the $L$ number of layers. At each layer $l$, the node embedding is updated as follows:

$$\mathbf{h}_v^{l+1} = \text{UPD}\left(\mathbf{h}_v^l, \text{AGG}\left(\left\{\text{MSG}(\mathbf{h}_v^l, \mathbf{h}_u^l) \mid \mathbf{A}_{uv} = 1\right\}\right)\right), \tag{1}$$

where $\mathbf{h}_v^0 = \mathbf{X}_v$ is the feature vector of node $v$, and $\mathbf{h}_v^l$ represents the node embedding at the layer $l$. The update function UPD, aggregation operation AGG, and message function MSG define the architecture of a GNN. For instance, Graph Convolutional Networks (Kipf & Welling, 2017) use an identity message function, mean aggregation, and a weighted update. A GNN model $\mathcal{M}$ performs graph classification by passing the graph embeddings $\mathbf{h}_G^L$ to a fully connected layer followed by a softmax function. Here, $\mathbf{h}_G^L$ is commonly computed by taking the mean of all node embeddings in the graph $\mathbf{h}_G^L := \text{mean}(\mathbf{h}_v^L \mid v \in V_G)$ through the operation `global_mean_pooling`.

**Node Classification.**   For node classification, the final embeddings are passed directly through a softmax function for individual label prediction. This approach omits the global pooling operation.

### 2.2   FIRST-ORDER LOGICAL RULES FOR GNN INTERPRETABILITY

First-order logic (FOL) is highly interpretable to humans, making it an excellent tool for explaining the behaviour of neural networks (Zhang et al., 2021). In this paper, our proposed framework, LOGICXGNN, aims to elucidate the inner decision-making process of a GNN $M$ using a *Disjunctive Normal Form (DNF)* formula $\phi_M$. The formula $\phi_M$ is a logical expression that can be described as a disjunction of conjunctions (OR of ANDs) over a set of predicates $P$, where each $p_j$ represents a property defined on the graph structure $\mathbf{A}$ and input features $\mathbf{X}$. Importantly, $\phi_M$ incorporates the *universal quantifier* ($\forall$), providing a global explanation that is specific to a class of instances.

While this approach looks promising, generating a DNF formula $\phi_M$ that faithfully explains the original GNN $M$ remains challenging. Specifically, we must address the following key questions:

1. How to define predicates $P$ that *reliably capture genuine structural patterns in the dataset, rather than just abstract symbols that lack effective grounding?*

2. How to derive *faithful* logical rules $\phi_M$ over $P$ that explains the GNN's predictions?

3. Can we design an approach that is both efficient (with minimal computational overhead) and generalizable to different tasks and GNN architectures?

## 3   THE LOGICXGNN FRAMEWORK ($\phi_M$)

### 3.1   IDENTIFYING HIDDEN PREDICATES $P$ FOR $\phi_M$

We begin by addressing the identification of hidden predicates for graph classification tasks. As discussed earlier, the desired predicates $P$ should capture commonly shared patterns in both graph structures $\mathbf{A}$ and hidden embeddings $\mathbf{h}^L$ across a set of instances in the context of GNNs. While graph

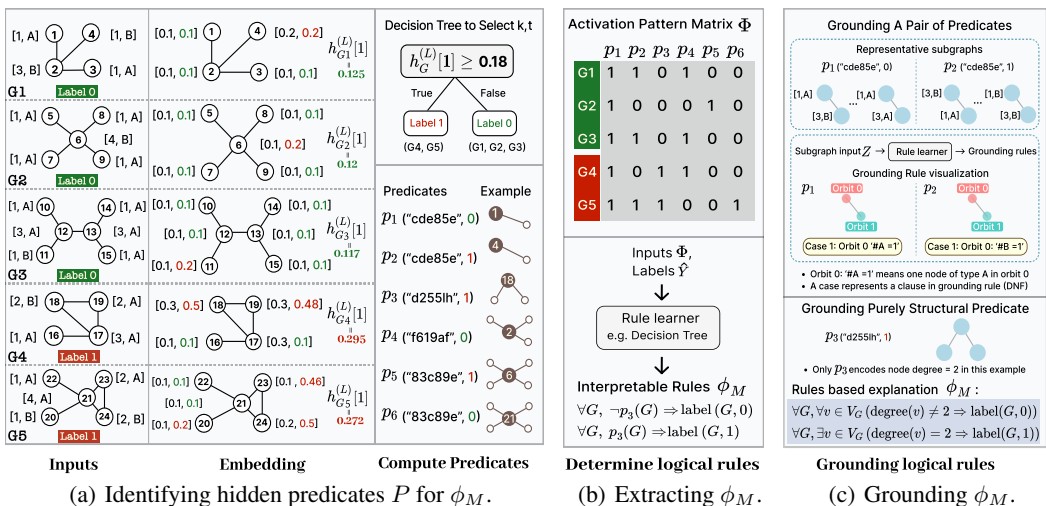

Figure 2: An overview of the LOGICXGNN framework, which involves identifying hidden predicates, extracting rules, and grounding these rules in the input space for interpretability.

structure information can be encoded into hidden embeddings, it often becomes indistinguishable due to oversmoothing during the message-passing process (Li et al., 2018; Xu et al., 2019).

The core of our approach is to explicitly model the recurring structural patterns that a GNN uses for computation. After $L$ layers of message passing in a GNN, the receptive field of a node $v$ is the subgraph induced by its $L$-hop neighborhood. Our key insight is that nodes with structurally identical (isomorphic) receptive fields share the same fundamental computational pattern. To systematically capture and compare these patterns, we use Weisfeiler–Lehman (WL) graph hashing to assign a unique identifier to each distinct receptive field topology.[1] This allows us to efficiently record and model these recurring structures. Formally, the structural pattern for a node $v$ is computed as follows:

$$Pattern_{\text{struct}}(v) = \text{Hash}\big(\text{ReceptiveField}(v, \mathbf{A}, L)\big). \tag{2}$$

Next, we discuss common patterns in the hidden embeddings. During GNN training, the hidden embeddings are optimized to differentiate between classes. Empirically, we find that a small subset of specific dimensions in the final-layer embeddings $\mathbf{h}_G^L$ is sufficient to distinguish instances from different classes using appropriate thresholds, often achieving accuracy comparable to the original GNN. Similar observations have been reported in Geng et al. (2025). In this work, we apply the decision tree algorithm to the collection of $\mathbf{h}_G^L$ from the training data to identify the most informative dimensions $K$ along with their corresponding thresholds $T$. Formally, this is expressed as:

$$\text{DecisionTree}(\{\mathbf{h}_G^L \mid G \in \mathcal{G}\}, \hat{Y}) \rightarrow (K, T) \tag{3}$$

where $\hat{Y}$ represents the prediction outcome of the GNN. We then leverage this information to construct embedding patterns at the node level, aligning with the definition of structural patterns. Recall that $\mathbf{h}_G^L := \text{mean}(\mathbf{h}_v^L \mid v \in V_G)$, so we broadcast $K$ and $T$ to each node embedding $\mathbf{h}_v^L$. Then, for an input node $v$, its embedding value $\mathbf{h}_v^L$ at each informative dimension $k \in K$ is compared against the corresponding threshold $T_k$. The result is then abstracted into binary states: 1 (activation) if the condition is met, and 0 (deactivation) otherwise. Formally, we have:

$$\mathcal{I}_k(\mathbf{h}_v^L) = 1 \text{ if } \mathbf{h}_v^L[k] \geq T_k, \text{ else } 0 \tag{4}$$

In summary, the embedding pattern contributed by a given node $v$ can be computed using the following function:

$$Pattern_{\text{emb}}(v) = \big[\mathcal{I}_1(\mathbf{h}_v^L), \mathcal{I}_2(\mathbf{h}_v^L), \dots, \mathcal{I}_K(\mathbf{h}_v^L)\big] \tag{5}$$

---

[1]We use WL hashing without node or edge features to capture the pure topological structure upon which the GNN's message-passing operates. This ensures our structural patterns have an expressiveness equivalent to that of standard GNNs.

Putting it together, we define the predicate function as $f(v) = (Pattern_{\text{struct}}(v), Pattern_{\text{emb}}(v))$. To identify the set of predicates, we iterate over each node $v \in \mathcal{V}$ in the training set, collect all $f(v)$, and transform them into a set $P$. In addition, when a node $v$ is evaluated against a predicate $p_j$, the evaluation $p_j(v)$ is true only if both the structural and embedding patterns from $f(v)$ match the predicate. To apply a predicate to a graph instance $G$, we override its definition as follows:

$$p_i(G) = 1 \text{ if } \exists v \in V_G, \ p_i(v) = 1, \quad p_i(G) = 0 \text{ if } \forall v \in V_G, \ p_i(v) = 0. \tag{6}$$

To better illustrate the process of identifying hidden predicates, we present a simple example in Figure 2(a). This scenario involves a binary graph classification task, a common setup in GNN applications. In this example, we have five input graphs, with each node characterized by two attributes: degree and type. The types are encoded as one-hot vectors. A GNN with a single message-passing layer is applied, generating a 2-dimensional embedding for each node (i.e., $d_L = 2$) and achieving 100% accuracy. As only one message-passing layer is used, structural patterns are extracted based on the nodes and their first-order neighbors.

Using decision trees, we identify the most informative dimension $k = 1$, and its corresponding threshold $t = 0.18$ from the graph embeddings. This threshold is then applied to the node embeddings to compute embedding patterns. As a result, six predicates are derived. Notably, $p_5$ ("83c89e", 1) and $p_6$ ("83c89e", 0) exhibit isomorphic structures, represented by identical hash strings, but differ in their activation patterns. Our predicates are therefore structurally grounded, as they capture concrete structural patterns from the training data, and model-faithful, since they are constructed by design to align with the GNN's predictions, $\hat{Y}$. This offers a significant advantage over prior methods, which often lack clear subgraph correspondence (Azzolin et al., 2023; Armgaan et al., 2024).

## 3.2 Determining the Logical Structure of $\phi_M$

The next task is to construct logical rules $\phi_M$ based on hidden predicates $P$ for each class, which serve as the explanation of the original GNN $M$. We process all training instances from class $c \in C$ that are correctly predicted by $M$, evaluating them against the predicates $P$ and recording their respective activation patterns. The results are stored in a binary matrix $\Phi_c$ for each class $c$, where the columns correspond to the predicates in $P$, and the rows represent the training instances. Specifically, an entry $\Phi_c[i, j] = 1$ denotes that the $j$-th instance exhibits the $i$-th predicate, while $\Phi_c[i, j] = 0$ indicates otherwise, as illustrated in Figure 2(b).

From a logical structure perspective, each row in $\Phi_c$ represents a logical rule that describes an instance of class $c$, expressed in conjunctive form using hidden predicates. For instance, in the simple binary classification task introduced earlier, $G_1$ corresponds to the column $(1, 1, 0, 1, 0, 0)$, which can be represented as $p_1 \wedge p_2 \wedge \neg p_3 \wedge p_4 \wedge \neg p_5 \wedge \neg p_6$. Here, we omit $G$ in $p_j(G)$ for brevity. To obtain a more compact set of explanation rules, we input $\Phi$ and $\hat{Y}$ into an off-the-shelf rule learner, such as decision trees or symbolic regression (Cranmer, 2023). In this work, we use decision trees for computational efficiency. The tree depth serves as a tunable parameter for controlling the complexity of the rules. For instance, in our simple GNN setting, the decision tree yields the following explanation rules $\phi_M$:

$$\forall G, \ \neg p_3(G) \Rightarrow \text{label}(G, 0), \quad \forall G, \ p_3(G) \Rightarrow \text{label}(G, 1). \tag{7}$$

## 3.3 Grounding $\phi_M$ into the Input Feature Space

The next challenge lies in grounding $\phi_M$. Prior work often simplifies this task by mapping each latent concept to a single representative subgraph, which can be poorly representative or even invalid (see our analysis of the root cause in the Appendix B.4). Such subgraphs can become especially meaningless when the input feature space $\mathbf{X}$ is continuous. To address these issues, LOGICXGNN goes beyond producing just individual explanation subgraphs. Moreover, it generates a set of generalized, fine-grained grounding rules that directly connect the hidden predicates $P$ to the input space $\mathbf{X}$. In particular, our predicate design explicitly integrates structural patterns, enabling both (i) the generation of diverse, representative subgraphs for each predicate and (ii) the construction of *structure-aware inputs* $\mathbf{Z}$ for inferring the grounding rules.

Recall that each $p_j$ can be represented by a collection of isomorphic subgraphs that activate $p_j$. Formally, given a graph $G = (V, E)$, we consider the action of the automorphism group $\text{Aut}(G)$ on its node set $V$. The orbit of a node $v \in V$ under this action is defined as

$$\text{Orb}(v) = \{u \in V \mid \exists \pi \in \text{Aut}(G) \text{ such that } \pi(v) = u\}. \tag{8}$$

Each orbit corresponds to an equivalence class of nodes that are structurally indistinguishable within $G$. To create a canonical representation, we partition the node set into these orbits and establish a consistent ordering using Algorithm 1 (Appendix D):

$$\mathcal{O}(G) = \{\text{Orb}(v_1), \text{Orb}(v_2), \ldots, \text{Orb}(v_k)\}, \tag{9}$$

where each $\text{Orb}(v_i)$ denotes the orbit of node $v_i$ under $\text{Aut}(G)$, and the ordering is deterministic and can be reliably reproduced across isomorphic subgraphs (see proofs in Appendix D).

**Definition 3.1** (Subgraph Input Feature $\mathbf{Z}$). The input features of nodes in a subgraph $G$ (represented by pattern $p_j$) are aggregated in a structure-aware manner according to the orbit ordering $\mathcal{O}(G)$:

$$\mathbf{Z}_G = \text{CONCAT}_{\text{Orb} \in \mathcal{O}(G)} \Big( \text{AGGREGATE}_{u \in \text{Orb}} \mathbf{X}_u \Big), \tag{10}$$

where AGGREGATE applies frequency encoding (mean encoding) for multi-node orbits with discrete (continuous) features, and the identity function for singleton orbits. Since each subgraph $G$ corresponds to the $L$-hop neighborhood of a central node $v$,[2] we adopt the notation $\mathbf{Z}_{v,L}$ in place of $\mathbf{Z}_G$ for convenience. For example, as shown in Figure 2(c), the subgraph input feature centered at node 1 is $\mathbf{Z}_{1,1} = (1, A, 3, B)$, which represents the concatenated features of nodes 1 and 2.

Once we obtain $\mathbf{Z}$, we can derive interpretable grounding rules that approximate the embedding pattern function $Pattern_{\text{emb}}(\cdot)$ encoded by the GNN. To address this, we leverage off-the-shelf rule learners; in this work, we utilize decision trees due to their computational efficiency and inherent interpretability. For predicates that exhibit isomorphic subgraph structures but distinct embedding patterns, we recast this problem as a supervised classification task, where each predicate $p_j$ is treated as a unique class label $j$. The training procedure constitutes a dataset for each predicate label $j$ by collecting the subgraph representations of all nodes $v$ that satisfy the predicate, formally defined as $\{\mathbf{Z}_{v,L} \mid p_j(v) = 1\}$. This process allows us to easily collect representative subgraphs, as shown in Figure 2(c). For example, the training data for $p_1$ (identified as ("cde85e", 0)) is $\{\mathbf{Z}_{1,1}, \mathbf{Z}_{3,1}, \ldots, \mathbf{Z}_{22,1}\}$, while the training data for $p_2$ (identified as ("cde85e", 1)) is $\{\mathbf{Z}_{4,1}, \mathbf{Z}_{11,1}, \mathbf{Z}_{20,1}\}$. Applying the decision tree then generates rules $\mathbf{Z}[1] \leq 0.5$ for $p_1$ and the opposite for $p_2$. Recall that $\mathbf{Z}[1]$, the first dimension of $\mathbf{Z}$, encodes the central node type. Therefore, we recognize that $p_1$ indicates that the central node is of type "A", while $p_2$ indicates type "B", conditioned on the structural pattern being "cde85e".

For purely structural predicates without direct embedding counterparts, explanations are grounded in the presence of their topological structures. Consider the predicate $p_3$, ("d255lh", 1), which activates when an input graph contains a subgraph isomorphic to the "d255lh" pattern, corresponding to a node with 2 edges, as illustrated in Figure 2(c). Since this is the only predicate in our GNN example that encodes this specific property, we can ground $\phi_M$ into the following interpretable logical rules:

$$\forall G, \forall v \in V_G \ (\text{degree}(v) \neq 2) \Rightarrow \text{label}(G, 0), \quad \forall G, \exists v \in V_G \ (\text{degree}(v) = 2) \Rightarrow \text{label}(G, 1). \tag{11}$$

However, such straightforward rules cannot always be derived in more complex scenarios. In general, the final rule-based explanation takes the form of logical rules over predicates, with predicates grounded either through grounding rules or representative subgraphs. More details about our grounding process, including the additional examples, handling of continuous features, and guidance on interpreting the grounding rule visualizations, are provided in Appendix D.

**Inference and Data-Grounded Fidelity.** During inference, a definitive prediction for a class is made if and only if the logical rule for that class is uniquely satisfied, as determined by evaluating each predicate on the input graph (Eq. 6). We then compute data-grounded fidelity ($Fid_\mathcal{D}$) as the *class-weighted* percentage of instances where this rule-based prediction exactly matches the original GNN's output. Note that a prediction is considered incorrect if it is ambiguous, which occurs when either no rule or multiple class rules are satisfied simultaneously. This issue is prevalent in prior methods, as shown with examples in Section 4.2. In contrast, our approach is guaranteed to avoid such ambiguity because its rules are derived from decision trees—a structure that inherently provides a unique classification for any given input. Further details on $Fid_\mathcal{D}$ and inference are in Appendix B.2.

**Remark on Node Classification.** Our framework naturally extends to node-level tasks by utilizing the predicate function $f(v) = (\text{Pattern}_{\text{struct}}(v), \text{Pattern}_{\text{emb}}(v))$ to encode class-informative signals

---

[2]Note that the central node $v$ can be treated as a singleton orbit, as adopted in our GNN example in Figure 2.

directly at the node level. Consequently, class-wise rules are obtained by aggregating the predicates associated with each class through a logical $\bigvee$ (OR) operation, bypassing the graph-level activation-matrix construction.

### 3.4 ANALYSIS

**Computational Complexity.** Our approach models message passing at each node to identify interpretable and reliable predicates. First, we extract activation patterns from pretrained GNNs and compute graph hashes over nodes' local neighborhoods. Hashing the $L$-hop neighborhood of a node takes approximately $O(L \cdot (\nu + \varepsilon))$ time, where $\nu$ and $\varepsilon$ denote the number of nodes and edges within the neighborhood. In practice, for well-structured and relatively sparse datasets, this hashing behaves nearly constant in runtime. Importantly, this step operates independently of the GNN's size, with overall complexity $O(|\mathcal{V}| \cdot L \cdot (\nu + \varepsilon))$. Second, we determine the logical structure by constructing a binary matrix of size (number of predicates) $\times$ (number of graphs), yielding a complexity of $O(|\mathcal{V}| \cdot |\mathcal{G}|)$. Finally, grounding each predicate requires constructing a dataset of representative subgraphs. Given that fitting a small decision tree is typically fast, often taking near-constant time in practice, this yields a complexity of $O(|\mathcal{V}|^2)$. A comprehensive analysis of the decision tree training overhead is provided in Appendix B.5. Empirical runtime results are reported in Table 1.

**Generalization Across Different GNN Architectures.** We show the theoretical generalizability of LOGICXGNN to any GNN architecture. First, we model stacked message-passing computations using hidden predicates (activation patterns and local subgraphs), an architecture-agnostic formulation. We then generate logical rules through binary matrix construction and decision tree analysis, maintaining architecture independence. Finally, we ground predicates by linking them to input features via decision trees, requiring no GNN-specific details. Empirical evidence is provided in Appendix C.4.

## 4 EVALUATION

In this section, we conduct extensive experimental evaluations on a broad collection of real-world benchmark datasets to investigate the following research questions:

1. How does $\phi_M$ perform compared to existing rule-based explanation methods across key metrics, including data-grounded fidelity, efficiency, and scalability?

2. How does $\phi_M$ improve explanation quality over existing approaches, and what are the key advantages of our generated explanations?

**Baselines.** Consistent with prior work (Armgaan et al., 2024), we restrict our comparison to global rule-based explanation methods, excluding local approaches such as GNNEXPLAINER (Ying et al., 2019) and (sub)graph generation-based approaches such as GNNINTERPRETER (Wang & Shen, 2023) due to their different scope. For evaluation, we compare our approach against state-of-the-art methods, GLGEXPLAINER (Azzolin et al., 2023) and GRAPHTRAIL (Armgaan et al., 2024).[3] Our primary evaluation metric is data-grounded fidelity, $Fid_{\mathcal{D}}$. Additional results on other metrics are provided in Appendix C.

Due to page limits, detailed descriptions of the datasets are provided in Appendix A, while the experimental setup, including GNN training and baseline implementations, can be found in Appendix B.

### 4.1 HOW EFFECTIVE IS $\phi_M$ AS A LOGICAL RULE-BASED EXPLANATION TOOL?

We report the data-grounded fidelity $Fid_D$ and runtime of our proposed approach $\phi_M$ and baseline methods on commonly used datasets for GNN explanation research, with results presented in Table 1. Results on large-scale real-world datasets are presented in Table 2. Notably, $\phi_M$ consistently outperforms both baselines by a substantial margin across all benchmarks. The performance gap arises because baseline explanation subgraphs are often poorly representative of the model's decision

---

[3]While GCNeuron (Xuanyuan et al., 2023) uses logic to describe individual neuron concepts, it relies on numerical importance scores rather than constructing explicit class-wise decision rules. Consequently, it lacks the translated logical rule-sets found in our method and the baselines, leading to its exclusion as a baseline.

Table 1: Data-grounded fidelity $Fid_{\mathcal{D}}$ (%) on the test datasets and runtime (in $10^3$ seconds) for various explanation methods. Results are reported over three random seeds. For each dataset, the highest fidelity and fastest runtime are highlighted in bold. "—" indicates no rules were learned.

| Method | BAMultiShapes | | BBBP | | Mutagenicity | | NCI1 | | IMDB | |
|---|---|---|---|---|---|---|---|---|---|---|
| | $Fid_{\mathcal{D}} \uparrow$ | $Time \downarrow$ | $Fid_{\mathcal{D}} \uparrow$ | $Time \downarrow$ | $Fid_{\mathcal{D}} \uparrow$ | $Time \downarrow$ | $Fid_{\mathcal{D}} \uparrow$ | $Time \downarrow$ | $Fid_{\mathcal{D}} \uparrow$ | $Time \downarrow$ |
| GLG | 31.09 ± 5.81 | 0.31 ± 0.02 | — | 0.36 ± 0.02 | 38.98 ± 3.01 | 0.73 ± 0.02 | 9.61 ± 7.76 | 0.88 ± 0.02 | 0.00 ± 0.00 | 0.33 ± 0.02 |
| GTRAIL | 79.82 ± 3.64 | 2.54 ± 0.12 | 50.00 ± 0.00 | 5.65 ± 0.12 | 65.93 ± 3.83 | 20.05 ± 1.12 | 60.04 ± 4.91 | 24.07 ± 1.12 | 35.35 ± 2.85 | 1.07 ± 0.02 |
| $\phi_M$ (Ours) | **82.67** ± 0.57 | **0.02** ± 0.00 | **85.32** ± 2.96 | **0.14** ± 0.01 | **81.36** ± 2.12 | **0.61** ± 0.10 | **73.81** ± 2.26 | **0.44** ± 0.00 | **74.16** ± 6.75 | **0.02** ± 0.00 |

Table 2: Data-grounded fidelity $Fid_D$ (%) on *large-scale* real-world datasets. `TO` indicates that the method did not complete within the allocated time limit of 12 hours. $\phi_M$ shows superior scalability compared to baseline methods.

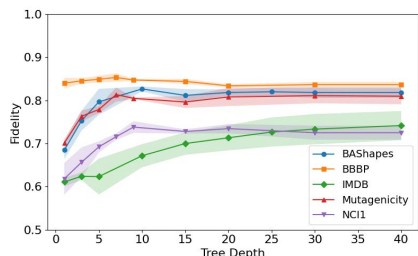

Figure 3: Impact of tree depth on $Fid_D$.

| Method | Reddit | | Twitch | | Github | |
|---|---|---|---|---|---|---|
| | $Fid_D \uparrow$ | $Time \downarrow$ | $Fid_D \uparrow$ | $Time \downarrow$ | $Fid_D \uparrow$ | $Time \downarrow$ |
| GLG | — | TO | — | TO | — | TO |
| GTrail | — | TO | — | TO | — | TO |
| $\phi_M$ (Ours) | 87.39 ± 0.59 | 4.04 ± 0.20 | 59.71 ± 0.92 | 7.27 ± 0.22 | 65.01 ± 2.59 | 12.17 ± 1.91 |

regions, or even fail to match any real graph instances in the dataset (e.g., GLGEXPLAINER yields 0% $Fid_D$ on IMDB). This performance gap is expected, as baselines often simplify grounding by mapping latent concepts to single representative subgraphs, which frequently results in poor representation or structural invalidity. We provide a detailed analysis of these grounding failures in Appendix B.4. In contrast, $\phi_M$ learns grounding rules that generalize well, explaining a large portion of unseen test data. We further analyze the quality of their explanations using concrete examples and additional utility metrics in Section 4.2. Another interesting observation is that $\phi_M$ can achieve relatively high fidelity even with simple rules, as shown in Figure 3. The tunable depth also gives users the flexibility to choose an appropriate trade-off between fidelity and rule complexity.

In terms of runtime performance, both baseline methods are fundamentally bottlenecked by their reliance on computationally expensive operations for each instance. For example, GLGEXPLAINER must invoke a separate local explainer, PGEXPLAINER (Luo et al., 2020a), for every graph, while GRAPHTRAIL requires numerous, costly GNN forward passes to process its computation trees. In contrast, $\phi_M$ employs highly efficient graph traversal algorithms and decision trees, yielding a dramatic speedup of one to two orders of magnitude (10–100×). This enables $\phi_M$ to demonstrate significantly better scalability on large-scale real-world datasets such as Reddit, Twitch, and Github (Rozemberczki et al., 2020), where both baselines time out, as shown in Table 2.

## 4.2 DOES $\phi_M$ PROVIDE BETTER EXPLANATIONS THAN EXISTING METHODS?

To assess the quality of our generated explanations, we conduct both qualitative and quantitative evaluations. Figures 4 and 5 provide a direct comparison of explanations from baseline approaches and $\phi_M$ on the datasets Mutagenicity and BBBP (Wu et al., 2018). Here, we use simplified rules for the baselines and $\phi_M$ (we set the tree depth for $\phi_M$ to 2 to achieve comparable rule complexity) for clearer visualization. This does not affect the nature of all methods. We choose these molecular datasets as they represent a domain that demands explanations with real-world scientific utility rather than mere subjective interpretability. An extended set of explanations is provided in Appendix E.3.

We identify several issues with both baselines. First, GLGEXPLAINER often generates conflicting rules. For example, in its rules for Mutagenicity, $x_{3333}$ is a subgraph of $x_{99}$, yet they correspond to different classes. This creates ambiguity because both class rules can be simultaneously satisfied. Consequently, GLGEXPLAINER reports a very low $Fid_{\mathcal{D}}$ of around 38.98%, as shown in Table 1. In some cases, it fails to yield any rules, as in BBBP. On the other hand, GRAPHTRAIL consistently produces chemically invalid motifs, such as $x_{149}$ and $x_{40}$ in BBBP. Moreover, since it generates only unilateral rules, all class 0 instances are trivially explained correctly, as they simply do not match these invalid subgraphs. Although GRAPHTRAIL appears to achieve a higher $Fid_{\mathcal{D}}$ than GLGEXPLAINER, such explanations remain largely meaningless to end users. In contrast, $\phi_M$ not

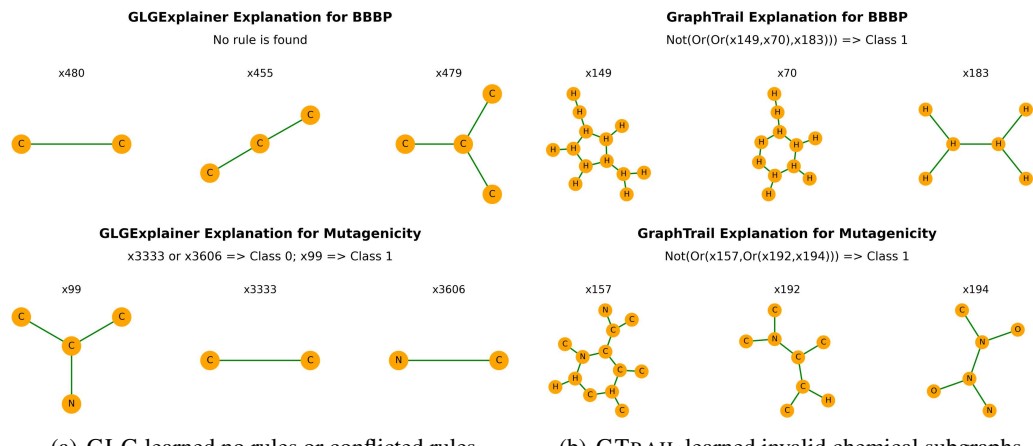

(a) GLG learned no rules or conflicted rules.

(b) GTRAIL learned invalid chemical subgraphs.

Figure 4: Baselines' explanations exhibit conflicting rules and chemically invalid subgraphs.

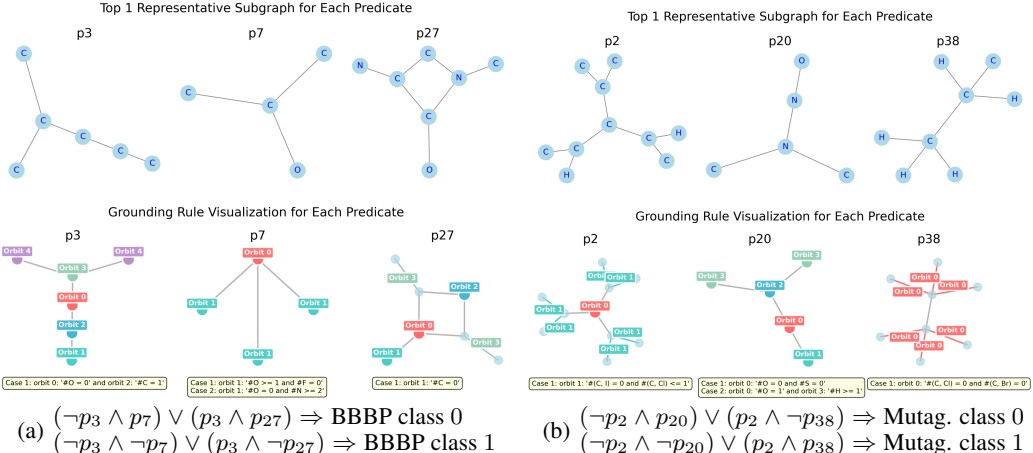

(a) $(\neg p_3 \wedge p_7) \vee (p_3 \wedge p_{27}) \Rightarrow$ BBBP class 0
$(\neg p_3 \wedge \neg p_7) \vee (p_3 \wedge \neg p_{27}) \Rightarrow$ BBBP class 1

(b) $(\neg p_2 \wedge p_{20}) \vee (p_2 \wedge \neg p_{38}) \Rightarrow$ Mutag. class 0
$(\neg p_2 \wedge \neg p_{20}) \vee (p_2 \wedge p_{38}) \Rightarrow$ Mutag. class 1

Figure 5: Besides representative subgraphs, our approach $\phi_M$ also provides general grounding rules for each predicate, effectively capturing more of the model's behavior, thereby achieving high fidelity.

only generates accurate representative subgraphs but also provides detailed general grounding rules for each predicate, resulting in a significantly higher $Fid_{\mathcal{D}}$ than the baselines. Moreover, our final explanations are expressed in DNF form, offering better readability than those of GRAPHTRAIL.

To complement this qualitative analysis, we quantitatively evaluate the generated explanations using a set of objective metrics that reflect their practical utility for end users: (1) *Coverage:* The proportion of target-class instances where the rule-based prediction remains correct when restricted to only valid subgraph patterns (i.e., after removing all invalid patterns). (2) *Stability:* The consistency of explanation subgraphs across multiple runs. (3) *Validity:* The proportion of explanation subgraphs

Table 3: *Coverage* of constructive explanations.

| Method | Mutagenicity | | BBBP | |
|---|---|---|---|---|
| | Class 0 (%) | Class 1 (%) | Class 0 (%) | Class 1 (%) |
| GLG | $6.11 \pm 4.17$ | $75.03 \pm 7.12$ | — | — |
| GTRAIL | $0.00 \pm 0.00$ | $78.48 \pm 1.04$ | $0.00 \pm 0.00$ | $0.00 \pm 0.00$ |
| $\phi_M$ (Ours) | $\mathbf{80.06} \pm 3.19$ | $\mathbf{82.58} \pm 1.35$ | $\mathbf{47.86} \pm 4.85$ | $\mathbf{98.16} \pm 0.57$ |

Table 4: *Stability* and *Validity*.

| Method | Stability(%) | | Validity(%) | |
|---|---|---|---|---|
| | Mutag. | BBBP | Mutag. | BBBP |
| GLG | 40.00 | — | $\mathbf{100.00} \pm 0.00$ | — |
| GTRAIL | 37.50 | 20.00 | $61.90 \pm 6.73$ | $0.00 \pm 0.00$ |
| $\phi_M$ (Ours) | $\mathbf{66.67}$ | $\mathbf{60.00}$ | $\mathbf{100.00} \pm 0.00$ | $\mathbf{100.00} \pm 0.00$ |

corresponding to valid chemical fragments in the dataset. Additional details on these metrics are provided in Appendix E.1. All results are computed and reported over three seeds in Tables 3 and 4.

Note that our approach consistently outperforms all baselines across these metrics. The high coverage indicates that our method provides meaningful explanations to more instances, while higher stability suggests more reliable, reproducible explanations. The 100% validity scores further confirm that our explanations correspond to chemically meaningful substructures, making them interpretable and trustworthy for domain experts. Appendix E.2 provides additional analyses, which corroborate our qualitative findings and validate the effectiveness of our approach for high-quality graph explanations.

**Validation on Synthetic Benchmarks.** We evaluate $\phi_M$'s rule-learning capability using the BA-MultiShapes dataset. Despite substantial noise in the underlying graph structures, our method accurately recovers the governing logical rules. Specifically, the ground-truth rule for Class 1 is the Disjunctive Normal Form: $(H \wedge W) \vee (H \wedge G) \vee (W \wedge G) \Rightarrow$ Class 1, where $H, W$, and $G$ represent House, Wheel, and Grid motifs. As shown in Figure 12 (Appendix E.3), our model extracts clauses mapping directly to these logical components; for instance, at rule depth 5, clauses $(p_{52} \wedge p_{55})$ and $(p_{280} \wedge p_{113})$ correspond to $(H \wedge W)$ and $(W \wedge G)$. The complete rule is fully recovered at depth 10. In contrast, state-of-the-art baselines, including GLGEXPLAINER and GRAPHTRAIL, fail to extract comparable rules on this benchmark (see Figure 5 in (Armgaan et al., 2024)).

## 5 RELATED WORK

Explainability methods for Graph Neural Networks (GNNs) can be broadly categorized into local and global approaches. A significant portion of prior work has focused on local explanations, which provide input attribution scores for a single prediction (Pope et al., 2019; Ying et al., 2019; Luo et al., 2020b; Vu & Thai, 2020; Lucic et al., 2022; Tan et al., 2022). These methods identify the most critical nodes and edges for a given decision, analogous to attribution techniques like Grad-CAM (Selvaraju et al., 2017) used in computer vision. In contrast, global explanations aim to capture the model's overall behavior, primarily through two strategies. (Sub)graph generation-base methods seek to identify representative graph patterns that are highly indicative of a particular class (Yuan et al., 2020; Wang & Shen, 2023; Xuanyuan et al., 2023; Wang & Shen, 2024; Saha & Bandyopadhyay, 2024; Lv & Chen, 2023; Yu & Gao, 2025). Logical rule-based methods, however, offer more expressive and human-readable explanations by using subgraphs as interpretable concepts within a formal logical formula. Our proposed method, LOGICXGNN, operates within this advanced domain of global rule-based explanations, aiming to generate precise and interpretable rules that clearly describe a GNN's decision-making process. Another related line of work involves self-explainable GNNs, which aim to develop model architectures that are inherently interpretable by design (Dai & Wang, 2021; Liu et al., 2025; Ragno et al., 2022). These methods are not directly compared in our work as they address a different goal, building interpretable models from scratch, whereas our focus is on providing post-hoc explanations for any pre-trained GNN. We believe that generalizing our rule-based framework to the domain of self-explainable models is a promising direction for future research.

## 6 CONCLUSION

In this work, we identify a fundamental limitation in existing rule-based explanation methods for GNNs: they optimize and evaluate fidelity in an intermediate, uninterpretable concept space while neglecting the grounding quality of final subgraph explanations presented to end users. This disconnect undermines both usability and trustworthiness, as methods often produce explanations that appear highly faithful yet fail to reflect concrete patterns in the data. To address this critical gap, we propose LOGICXGNN, a novel framework that constructs explanation rules over reliable predicates designed to preserve structural patterns inherent in GNN's message-passing mechanism. Our approach enables effective grounding that produces representative subgraphs and learns generalizable grounding rules. LOGICXGNN achieves an average improvement of over 20% in data-grounded fidelity $Fid_{\mathcal{D}}$ while delivering 10–100× computational speedup compared to existing methods. Comprehensive evaluation across coverage, stability, and validity metrics confirms that LOGICXGNN produces explanations with genuine practical utility, significantly advancing trustworthy GNN explainability. For future work, we aim to enhance explanation interpretability by integrating domain knowledge, particularly in biochemistry, to uncover novel structure-activity relationships within complex molecular datasets.

## ACKNOWLEDGMENTS

This work was supported, in part, by Individual Discovery Grants from the Natural Sciences and Engineering Research Council (NSERC) of Canada and the Canada CIFAR AI Chair Program. This research was also supported by the ProML project, a joint initiative funded by NSERC and the French National Research Agency (ANR) under the reference ANR-25-CE23-6715.

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

## A  DATASET DETAILS

We evaluate our approach on a diverse set of graph classification benchmarks commonly used in GNN explanation research. Table 5 summarizes the statistics of these datasets.

- **Molecular Graphs:** Mutagenicity (Debnath et al., 1991), NCI1 (Wale et al., 2008), and BBBP (Wu et al., 2018) are molecular datasets where nodes represent atoms and edges represent chemical bonds. In NCI1, each graph is labeled according to its anticancer activity. Mutagenicity contains compounds labeled based on their mutagenic effect on the Gram-negative bacterium (Label 0 indicates mutagenic). BBBP labels molecules by their ability to penetrate the blood-brain barrier.
- **Synthetic Graphs:** BAMultiShapes (BAShapes) consists of 1,000 Barabási-Albert (BA) graphs with randomly placed network motifs such as house, grid, and wheel structures (Ying et al., 2019). Class 0 contains plain BA graphs or those with one or more motifs, while Class 1 contains graphs enriched with two motif combinations.
- **Social Graphs:** IMDB-BINARY (IMDB) represents social networks where each graph corresponds to a movie; nodes are actors and edges indicate co-appearances in scenes (Morris et al., 2020).

Table 5: Statistics of standard graph datasets.

|  | BAMultiShapes | Mutagenicity | BBBP | NCI1 | IMDB |
|---|---|---|---|---|---|
| #Graphs | 1,000 | 4,337 | 2,050 | 4,110 | 1,000 |
| Avg. $|\mathcal{V}|$ | 40 | 30.32 | 23.9 | 29.87 | 19.8 |
| Avg. $|\mathcal{E}|$ | 87.00 | 30.77 | 51.6 | 32.30 | 193.1 |
| #Node features | 10 | 14 | 9 | 37 | 0 |

To assess scalability, we also benchmark our approach and baselines on large-scale, real-world datasets from Rozemberczki et al. (2020): Reddit Threads, Twitch Egos, and GitHub Stargazers. Table 6 summarizes their statistics.

- **Reddit Threads** (Reddit): Labeled as discussion-based or non-discussion-based. The task is to predict whether a thread is discussion-oriented.
- **Twitch Egos** (Twitch): Ego networks of Twitch users. The task is to predict whether the central gamer plays a single game or multiple games.
- **GitHub Stargazers** (GitHub): Social networks of developers who starred popular machine learning or web development repositories. The task is to classify whether a social network belongs to a web or machine learning repository.

Table 6: Statistics of Graph Datasets: Nodes, Density, and Diameter

| Dataset | Graphs | Nodes | | Density | | Diameter | |
|---|---|---|---|---|---|---|---|
|  |  | Min | Max | Min | Max | Min | Max |
| Reddit | 203,088 | 11 | 197 | 0.021 | 0.382 | 2 | 27 |
| Twitch | 127,094 | 14 | 452 | 0.038 | 0.967 | 1 | 12 |
| GitHub | 12,725 | 10 | 957 | 0.003 | 0.561 | 2 | 18 |

## B  EXPERIMENTAL SETUP AND REPLICATE BASELINES

### B.1  EXPERIMENTAL SETUP

All experiments are conducted on an Ubuntu 22.04 LTS machine with 128 GB RAM and an AMD EPYC™ 7532 processor. Each dataset is split into training and testing sets using an 80/20 ratio, and

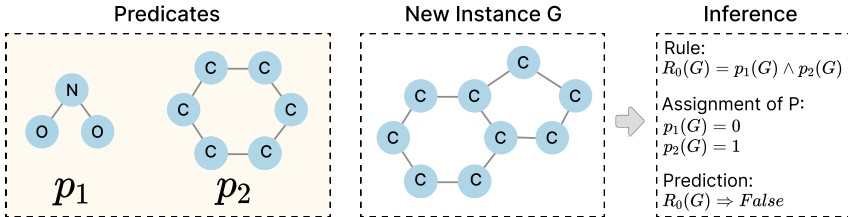

Figure 6: The example for inference.

all experiments are repeated with three different random seeds to ensure robustness. The seeds affect the entire pipeline, including GNN training. Although our method itself is deterministic, we evaluate it on GNNs trained with different seeds to ensure fair comparison and to assess robustness under natural variations, following standard XAI practice.

The default GNN architecture is GCN (Kipf & Welling, 2017) for all benchmarks. To demonstrate the model-agnostic nature of LOGICXGNN, we additionally benchmark against multiple GNN architectures, including 2-layer GraphSAGE (Hamilton et al., 2018), 3-layer GIN (Xu et al., 2019), and 2-layer GAT (Velickovic et al., 2018). Results are reported in Table 13. For GNN training, we use the Adam optimizer with a learning rate of 0.005. Each GNN is trained for up to 500 epochs with early stopping after a 100-epoch warm-up if validation accuracy does not improve for 50 consecutive epochs. *All explanation methods are trained and learned on the same training data as the base GNNs, and their performance is evaluated on the test splits. The main metric is data-grounded fidelity $Fid_{\mathcal{D}}$.*

As for our approach, LOGICXGNN, we employ the CART algorithm for all decision trees used in LOGICXGNN. To compute the predicates, we first select the top K most informative dimensions from the decision tree that achieves at least 95% accuracy (Eq. 3). We then use these dimensions to generate activation embeddings for each predicate and apply Weisfeiler-Lehman (WL) hashing to capture their topological structure. We report the best results across different tree depths for the final rule-based explanations. For each explanation method, we allocate a time limit of 12 hours, excluding the training time of the GNNs.

## B.2 INFERENCE WITH RULE-BASED EXPLANATIONS ON GRAPH SPACE

Rule-based GNN explanations operate by evaluating a set of logical formulas defined over an input graph. During inference, these formulas are applied to an input graph $G$ to generate a prediction. Typically, each symbol—i.e., a predicate or concept—in a formula evaluates to `True` if its corresponding subgraph is present in $G$ (i.e., an isomorphic subgraph is detected).

To provide a concrete example, to make a final prediction for an input graph $G$, we define $R_c$ as the logical rule for a given class $c$. The graph $G$ is classified as class $c$ if the rule $R_c$ evaluates to `True` based on the values of the predicates (symbols) appearing in that formula. For instance, consider a GNN trained to predict mutagenicity, with a rule for the class "Mutagenic" ($R_0$). The explanation method might produce the following rule based on chemical substructure predicates:

- **Predicate $p_1$**: `True` if the input graph contains a nitro group ($NO_2$).
- **Predicate $p_2$**: `True` if the input graph contains a 6-carbon ring.
- **Rule $R_0$ (Mutagenic)**: $p_1 \wedge p_2$

Now, consider an input molecule $G$ that contains a 6-carbon ring but no nitro group. Predicate $p_1(G)$ evaluates to `False`, while predicate $p_2(G)$ evaluates to `True`. The rule $R_0(G) = p_1(G) \wedge p_2(G)$ therefore becomes `False` $\wedge$ `True`, which evaluates to `False`. Consequently, the input graph $G$ is **not** classified as "Mutagenic" according to this rule, as illustrated in Figure 6.

We use an efficient subgraph isomorphism matching algorithm (`igraph.subisomorphic_vf2`) to perform subgraph matching and evaluate the rules for the baseline approaches. A timeout of 10 minutes is applied to each instance (only a few cases in our experiments reached this limit). Final results are computed over all instances, excluding those that timed out.

Note that in our approach, LOGICXGNN, a predicate evaluates to `True` under either of two conditions: (1) when a matching subgraph is found in $G$, or (2) when a subgraph in $G$ satisfies the corresponding grounding rule over its features $Z$, given that it matches the structural pattern. We adopt condition (2) as it provides a more comprehensive evaluation framework, incorporating both structural and feature-based constraints and going beyond simple pattern matching to a rule-driven assessment. This design allows LOGICXGNN to identify functionally equivalent subgraphs by leveraging the GNN's learned representations, offering more accurate explanations and better generalization than baselines, which rely solely on purely structural matching.

**Computing Data-Grounded Fidelity $Fid_{\mathcal{D}}$.** Following the rule-inference paradigm introduced earlier, the ground truth for computing data-grounded fidelity ($Fid_{\mathcal{D}}$) is taken to be the GNN's own predictions. Consider a binary classification setting where the GNN label is either 0 or 1. The rule-based explainer produces a tuple of outputs, for example $(\text{rule}_0(G), \text{rule}_1(G))$, indicating whether the corresponding class rule evaluates to `True`. For example, in the binary classification setting, the possible outputs are:

$$(0,0), \quad (0,1), \quad (1,0), \quad (1,1).$$

When computing $Fid_{\mathcal{D}}$, the cases $(0,0)$ and $(1,1)$ are treated as `False`, since they indicate either multiple or no classes are satisfied, making the prediction ambiguous. A prediction is counted as correct only when there is an exact match between the GNN label and the rule-based prediction; for example, if the GNN label is 0, then the rule output must be $(1,0)$ for it to be considered correct. The same computation scheme is also applied to other metrics used in this paper, such as accuracy, recall, precision, and F1 score. Finally, to address class imbalance, we incorporate a weighting scheme so that underrepresented classes are not penalized disproportionately in the fidelity computation.

Formally, let $y_{\text{GNN}}(G)$ denote the GNN prediction for graph $G$, and let $\hat{y}_{\text{rule}}(G)$ denote the prediction of the rule-based explainer (defined only when exactly one class rule fires). With class weights $w_c > 0$, the data-grounded fidelity is defined as:

$$Fid_{\mathcal{D}} = \frac{\sum_{G \in \mathcal{D}_{\text{test}}} w_{\, y_{\text{GNN}}(G)} \cdot \mathbb{I}(\hat{y}_{\text{rule}}(G) = y_{\text{GNN}}(G))}{\sum_{G \in \mathcal{D}_{\text{test}}} w_{\, y_{\text{GNN}}(G)}}, \tag{12}$$

where $\mathbb{I}(\cdot)$ is the indicator function. This weighted formulation ensures that classes with fewer samples contribute proportionally, making $Fid_{\mathcal{D}}$ a fair measure of agreement between the GNN and the rule-based explainer.

### B.3 REPRODUCTION OF BASELINE APPROACHES

**Reproduction of GLGEXPLAINER.** We conducted experiments on GLGEXPLAINER (Azzolin et al., 2023) using their official GitHub repository. Following the original paper, we adopted the default hyperparameter settings and consulted with the original authors to verify our understanding and methodology, ensuring a fair and faithful evaluation.

During our study on this baseline, we made several important observations:

1. **Incomplete Implementation:** The public codebase for GLGEXPLAINER relies on pre-computed local explanations from PGEXPLAINER (Luo et al., 2020a) but omits the code for generating them. This prevents the method from being applied to new datasets out of the box. Following the authors' guidance, we integrated the official PGEXPLAINER repository and performed the necessary hyperparameter tuning to generate these essential local explanations running properly for our experiments.

2. **Instability:** We observed that the explanation quality of both PGEXPLAINER and GLGEXPLAINER is highly sensitive to hyperparameter choices and random seeds. The original authors confirmed they also faced challenges in consistently reproducing results, attributing it to "high stochasticity" in the training process. This inherent instability means that explanations can differ substantially across runs, affecting direct reproducibility—a limitation they themselves highlight in the paper.

3. **Reliance on Domain Knowledge:** The method requires external domain knowledge to derive concept representations from local explanations. To replicate the original authors' setup, we used concepts learned from our own approach to supply this necessary domain knowledge to GLGEXPLAINER.

In summary, our reproduced results are largely consistent with those reported in the original paper, as shown in Table 8. Specifically, the reproduced fidelity in the intermediate abstract concept space aligns closely with the results reported in the original paper (Azzolin et al., 2023) and in subsequent work (Armgaan et al., 2024). Moreover, our experiments on GRAPHTRAIL with new datasets were carefully conducted under the guidance of the original authors. Taken together, we are confident that our experimental setup is fair and that our results on GRAPHTRAIL constitute a valid comparison.

**Reproduction of GRAPHTRAIL.**  We conducted experiments on GRAPHTRAIL (Armgaan et al., 2024) using their official GitHub repository. Following the original paper, we adopted the default hyperparameter settings and consulted with the original authors to verify our understanding and methodology, ensuring a fair and faithful evaluation.

During our study on this baseline, we made several important observations:

1. **Instability:** We observed that the explanations vary significantly across different seeds. The authors also acknowledged this issue. They confirmed that final subgraph explanations may indeed vary from seed to seed, although the fidelity values remain stable. This highlights that GraphTrail's symbolic rules are not deterministic and depend on stochastic elements of the pipeline.

2. **Chemically Invalid Motifs:** We observed many invalid subgraph explanations for molecular datasets. The authors admitted that invalid-looking subgraphs (e.g., a hydrogen atom appearing in a ring, which is chemically impossible) can occur. In the paper, they stated that subgraphs were manually redrawn to avoid such errors. This admission suggests that the published qualitative results required manual intervention and that the current pipeline cannot guarantee chemically valid visualizations.

3. **Fidelity Concerns:** The authors claimed that issues such as invalid subgraphs do not affect fidelity, since fidelity is based on c-trees. However, this also means that fidelity does not fully capture the validity or interpretability of the symbolic rules. In practice, fidelity values may be correct while the extracted rules remain trivial or domain-invalid.

In summary, our reproduced results are largely consistent with those reported in the original paper, as shown in Table 8. In particular, the reproduced fidelity in the intermediate abstract concept space aligns closely with the original findings (Armgaan et al., 2024). Taken together with our direct communication with the original authors, we are confident that our experimental setup is fair and that our results on GRAPHTRAIL provide a valid comparison.

Moreover, the issue of *chemically invalid motifs*, which the original authors themselves acknowledged, provides further motivation for the development of our proposed LOGICXGNN. By explicitly addressing the unreliable grounding of baseline methods, LOGICXGNN ensures that the learned explanations are not only faithful but also scientifically valid and interpretable.

### B.4  THE INEFFECTIVE GROUNDING ISSUE OF BASELINE APPROACHES

**The grounding issue of GLGEXPLAINER.**  While the subgraphs produced by GLGEXPLAINER are structurally valid, their grounding often results in explanations that are *unrepresentative or trivial*. The primary issue lies in its two-stage, post-hoc process, which first clusters a large set of local explanations into abstract concepts and then selects a representative subgraph for each.

As the code in Figure 7 illustrates, the method simply visualizes the top examples that best match a learned prototype after the concepts have already been formed. The critical weakness is that if the initial clustering groups dissimilar or noisy local explanations together, the resulting "concept" becomes incoherent. Consequently, the final representative subgraph, though a valid member of the cluster, may only be a trivial or poorly representative example, leading to an unfaithful explanation of the model's behavior.

**The grounding issue of GRAPHTRAIL.**  During our analysis, we identified a critical issue with how GRAPHTRAIL grounds its explanations by generating subgraphs from computation trees, a problem the original authors acknowledge as a bug. Upon inspection, we found that the method *reconstructs invalid subgraphs by mismatching node attributes with the underlying graph structure.*

materialize prototypes

```
In [16]:  # change assign function to a non-discrete one just to compute distance between local expls. and prototypes
          # useful to show the materialization of prototypes based on distance
          best_expl.hyper["assign_func"] = "sim"

          x_train , emb , concepts_assignement , y_train_1h , le_classes , le_idxs , belonging = best_expl.get_concept_vect

          best_expl.hyper["assign_func"] = "discrete"

          proto_names = {
              0: "Others",
              1: "$NO_2$",
          }
          torch.manual_seed(42)
          fig = plt.figure(figsize=(17,4))
          n = 0
          for p in range(best_expl.hyper["num_prototypes"]):
              idxs = le_idxs[concepts_assignement.argmax(-1) == p]
              idxs = idxs[torch.randperm(len(idxs))] # for random examples
              sa = concepts_assignement[concepts_assignement.argmax(-1) == p]
              idxs = idxs[torch.argsort(sa[:, p], descending=True)]

              for ex in range(5):
                  n += 1
                  plt.subplot(best_expl.hyper["num_prototypes"],5,n)
                  utils.plot_molecule(dataset_train[int(idxs[ex])], composite_plot=True)

          for p in range(best_expl.hyper["num_prototypes"]):
              plt.subplot(best_expl.hyper["num_prototypes"],5,5*p + 1)
              plt.ylabel(f"$P_{p}$\n {proto_names[p]}", size=25, rotation="horizontal", labelpad=50)

          plt.show()
```

Figure 7: Code from GLGEXPLAINER for selecting representative subgraphs. This post-hoc selection can yield unrepresentative examples if the underlying concept cluster is poorly defined or contains noisy local explanations.

This error originates in the `utils.dfs` function, shown in Figure 8. The function attempts to merge two different graph representations: `ctree` (containing rich attributes like atom types) and `ctree_id` (using simple integer IDs). It operates on the flawed assumption that the nodes in both graphs are identically ordered, mapping them by their list position rather than a stable identifier. Crucially, the original author's comment in the code, `! Incorrect`, explicitly acknowledges this flawed premise. Consequently, the subgraphs produced by this function are often structurally invalid, undermining their reliability as faithful explanations.

```
 96  ∨  def dfs(ctree, ctree_id, node_mapping=None):
 97          """
 98          ! Incorrect
 99          The ctree_id code is generated by writing down the node id as the canonical label of ctree
100          is generated. Hence, the node order between the two is preserved. Therefore, we can map
101          ctree's node attriburtes to ctree_id's node attributes.
102          """
103          G = nx.Graph()
104          for i in range(len(ctree_id.nodes)):
105              if node_mapping is not None:
106                  attr = node_mapping[ctree.nodes[i]['attr']]
107              else:
108                  attr = ctree.nodes[i]['attr']
109              G.add_node(ctree_id.nodes[i]['attr'], attr=attr)
110          for e in ctree_id.edges:
111              src, dest = e
112              src = ctree_id.nodes[src]['attr']
113              dest = ctree_id.nodes[dest]['attr']
114              G.add_edge(src, dest, attr=ctree_id.edges[e]['attr'])
115          return G
```

Figure 8: The flawed `utils.dfs` function from the official GRAPHTRAIL repository. The original author's comment (`! Incorrect`) confirms the function's incorrect assumption about node ordering when reconstructing subgraphs.

**Why our approach $\phi_M$ achieves effective grounding?** The effectiveness of our grounding process for $\phi_M$ stems from its rigorous, data-driven foundation. Our method begins with a systematic *cataloging of all structural patterns* as they appear in the training data. This detailed "bookkeeping" ensures that every subgraph used for grounding is guaranteed to be both *structurally valid and highly representative*, as it is drawn directly from real instances.

Furthermore, $\phi_M$ moves beyond simply showing these examples. It learns formal *grounding rules* on top of this empirical collection, providing a precise, logical explanation for *why* a given structural pattern is important for the model's prediction. This combination of data-backed, representative

subgraphs and the formal rules that govern them provides a grounding that is both faithful to the data and deeply interpretable.

### B.5 ADDITIONAL RUNTIME COMPLEXITY ANALYSIS

The complexity analysis in the main paper simplifies the overhead of training decision trees due to space constraints. We clarify here the computational cost associated with training the grounding decision trees. For each predicate $p$, we construct a dataset of representative subgraphs and train a shallow decision tree on low-dimensional structural features. Standard CART-style training has a complexity of $O(d_p N_p \log N_p)$. In our implementation, the feature dimension $d_p$ is a small constant representing simple predicate-level statistics. Furthermore, the number of training examples $N_p \leq |\mathcal{V}|$ because each example corresponds to a grounded subgraph, and the total number of predicates is bounded by $O(|\mathcal{V}|)$.

Summing the cost across all predicates yields the total complexity:

$$\sum_p O(d_p N_p \log N_p) \leq \sum_{O(|\mathcal{V}|)} O(1 \cdot |\mathcal{V}| \log |\mathcal{V}|) = O(|\mathcal{V}|^2 \log |\mathcal{V}|). \tag{13}$$

This result is consistent with the $O(|\mathcal{V}|^2)$ grounding term reported in the main paper. The additional $\log |\mathcal{V}|$ factor is mild in practice; since the trees are intentionally kept shallow and $d_p$ is constant, this overhead remains negligible compared to the costs of subgraph extraction and predicate construction.

## C ADDITIONAL EVALUATION RESULTS

Table 7: Running record on three large datasets using 4 CPU cores.

| Dataset | Seed | $Fid_{\mathcal{D}}$ (%) | Mem. (GB) | Time | CPU usage |
|---|---|---|---|---|---|
| Reddit Threads | 0 | 86.82 | 36.1 | 1 h 06 min | 102 |
| Reddit Threads | 1 | 87.35 | 36.7 | 1 h 06 min | 103 |
| Reddit Threads | 2 | 87.99 | 36.7 | 1 h 10 min | 98 |
| Twitch-Egos | 0 | 57.49 | 69.2 | 2 h 02 min | 98 |
| Twitch-Egos | 1 | 57.81 | 69.5 | 2 h 13 min | 92 |
| Twitch-Egos | 2 | 55.01 | 69.2 | 2 h 18 min | 95 |
| Github Stargazers | 0 | 65.79 | 64.9 | 3 h 35 min | 80 |
| Github Stargazers | 1 | 59.49 | 84.6 | 3 h 43 min | 95 |
| Github Stargazers | 2 | 66.85 | 79.9 | 3 h 18 min | 97 |

### C.1 FIDELITY COMPARISON

We compare the original fidelity—computed in the intermediate, uninterpretable concept space and reported by the baselines—against their data-grounded fidelity ($Fid_{\mathcal{D}}$), computed in the final grounded form, in Table 8. Under this more rigorous metric, the performance of existing methods drops significantly, underscoring the need for explanations that are both faithful and genuinely interpretable.

### C.2 RUNNING RECORD ON THREE LARGE-SCALE BENCHMARKS

We report the running record of our approach, LOGICXGNN, on three large-scale benchmarks in Table 7. Our method is the only existing approach that can scale to this size with high efficiency, as indicated by its low and stable memory usage.

### C.3 ADDITIONAL METRICS ON EXPERIMENTS OVER FIVE COMMON DATASETS

To ensure robust and reliable comparisons, in addition to Table 1, we also report the weighted test accuracy, precision, recall, and F1-score, averaged across three runs, in Tables 9, 10, 11, and 12,

Table 8: Data-grounded fidelity $Fid_{\mathcal{D}}$ (%) in percentage and original fidelity $Fid$ of all baselines, averaged over three random seeds. For each dataset, the highest fidelity is highlighted in bold. Since our approach reports only $Fid_{\mathcal{D}}$, its $Fid$ entries are omitted and marked with "–". "—" indicates cases where no rules were learned.

| Method | BAShapes | | BBBP | | Mutagenicity | | NCI1 | | IMDB | |
| | $Fid$ | $Fid_{\mathcal{D}}$ | $Fid$ | $Fid_{\mathcal{D}}$ | $Fid$ | $Fid_{\mathcal{D}}$ | $Fid$ | $Fid_{\mathcal{D}}$ | $Fid$ | $Fid_{\mathcal{D}}$ |
|---|---|---|---|---|---|---|---|---|---|---|
| GLG | $57.50_{\pm 0.50}$ | $31.09_{\pm 5.81}$ | $52.50_{\pm 0.50}$ | — | $62.16_{\pm 2.19}$ | $38.98_{\pm 3.01}$ | $58.09_{\pm 2.51}$ | $9.61_{\pm 7.76}$ | $53.50_{\pm 0.50}$ | $0.00_{\pm 0.00}$ |
| G-Trail | $84.67_{\pm 4.77}$ | $79.82_{\pm 3.65}$ | $97.17_{\pm 0.89}$ | $50.00_{\pm 0.00}$ | $73.90_{\pm 1.49}$ | $65.93_{\pm 3.83}$ | $68.70_{\pm 0.93}$ | $60.04_{\pm 4.91}$ | $66.67_{\pm 10.93}$ | $35.35_{\pm 2.85}$ |
| $\phi_M$ (Ours) | – | $\mathbf{82.67}_{\pm 0.58}$ | – | $\mathbf{85.32}_{\pm 2.96}$ | – | $\mathbf{81.36}_{\pm 2.12}$ | – | $\mathbf{73.81}_{\pm 2.26}$ | – | $\mathbf{74.17}_{\pm 6.75}$ |

respectively. The highest scores are highlighted in bold. Note that precision, recall, and F1-score are computed against the GNN predictions, following the evaluation protocol used in GraphTrail Armgaan et al. (2024).

Table 9: Test accuracy of various explanation methods (%) on graph classification datasets.

| Method | BAShapes | BBBP | Mutagenicity | NCI1 | IMDB |
|---|---|---|---|---|---|
| GLG | $41.14 \pm 4.81$ | — | $38.37 \pm 0.46$ | $3.73 \pm 8.57$ | $0.00 \pm 0.00$ |
| G-Trail | $78.88 \pm 5.14$ | $55.10 \pm 3.82$ | $59.72 \pm 4.37$ | $56.69 \pm 1.46$ | $30.20 \pm 4.23$ |
| $\phi_M$ (Ours) | $\mathbf{82.67 \pm 0.57}$ | $\mathbf{78.06 \pm 4.28}$ | $\mathbf{68.69 \pm 2.01}$ | $\mathbf{63.63 \pm 2.93}$ | $\mathbf{74.16 \pm 6.75}$ |

Table 10: Weighted precision of various explanation methods (%) on graph classification datasets.

| Method | BAShapes | BBBP | Mutagenicity | NCI1 | IMDB |
|---|---|---|---|---|---|
| GLG | $28.85 \pm 5.49$ | — | $70.63 \pm 6.59$ | $67.64 \pm 4.45$ | $0.00 \pm 0.00$ |
| G-Trail | $80.05 \pm 6.48$ | $59.12 \pm 2.81$ | $64.45 \pm 1.14$ | $61.18 \pm 2.51$ | $39.60 \pm 9.66$ |
| $\phi_M$ (Ours) | $\mathbf{82.76 \pm 0.61}$ | $\mathbf{92.10 \pm 0.88}$ | $\mathbf{81.71 \pm 1.77}$ | $\mathbf{74.19 \pm 1.73}$ | $\mathbf{82.78 \pm 2.68}$ |

Table 11: Weighted recall of various explanation methods (%) on graph classification datasets.

| Method | BAShapes | BBBP | Mutagenicity | NCI1 | IMDB |
|---|---|---|---|---|---|
| GLG | $31.09 \pm 5.81$ | — | $38.98 \pm 3.01$ | $9.61 \pm 7.76$ | $0.00 \pm 0.00$ |
| G-Trail | $79.82 \pm 3.65$ | $50.00 \pm 0.00$ | $65.93 \pm 3.83$ | $60.04 \pm 4.91$ | $35.35 \pm 2.85$ |
| $\phi_M$ (Ours) | $\mathbf{82.67 \pm 0.57}$ | $\mathbf{85.32 \pm 2.96}$ | $\mathbf{81.36 \pm 2.12}$ | $\mathbf{73.81 \pm 2.26}$ | $\mathbf{74.16 \pm 6.75}$ |

Table 12: Weighted F1-score of various explanation methods (%) on graph classification datasets.

| Method | BAShapes | BBBP | Mutagenicity | NCI1 | IMDB |
|---|---|---|---|---|---|
| GLG | $31.77 \pm 3.46$ | — | $32.08 \pm 0.46$ | $11.66 \pm 8.29$ | $0.00 \pm 0.00$ |
| G-Trail | $78.26 \pm 5.15$ | $66.84 \pm 2.49$ | $57.53 \pm 5.82$ | $53.01 \pm 3.92$ | $32.45 \pm 1.94$ |
| $\phi_M$ (Ours) | $\mathbf{82.68 \pm 0.56}$ | $\mathbf{92.11 \pm 0.78}$ | $\mathbf{81.44 \pm 1.92}$ | $\mathbf{73.66 \pm 2.02}$ | $\mathbf{71.43 \pm 7.82}$ |

## C.4 Empirical evidence for generalizability across GNN architectures

Table 13 reports the best baseline fidelity (Base.) and the fidelity of our approach, $\phi_M$, across multiple GNN architectures. The table also includes the classification accuracy of the underlying GNN models ($M$) for reference. Note that the GNN architectures **GraphSAGE** and **GAT** achieve only 47.50% accuracy on the BAShapes dataset due to their limited expressive power. These results are consistent with the findings reported in Armgaan et al. (2024). $\phi_M$ consistently achieves high fidelity across all architectures and datasets, uniformly outperforming the baselines. This demonstrates both (1) the strong generalizability of $\phi_M$ across different GNNs and (2) its state-of-the-art explanatory fidelity.

Table 13: Fidelity comparison of LOGICXGNN against baseline methods *across multiple GNN architectures*. $M$ denotes the underlying model's classification accuracy (%); *Base.* denotes the best baseline fidelity (%) from the better of GLGEXPLAINER and GRAPHTRAIL; and $\phi_M$ denotes the fidelity of our method, LOGICXGNN (%).

| Dataset | GCN | | | GraphSAGE | | | GIN | | | GAT | | |
|---|---|---|---|---|---|---|---|---|---|---|---|---|
| | $M$ | *Base.* | $\phi_M$ | $M$ | *Base.* | $\phi_M$ | $M$ | *Base.* | $\phi_M$ | $M$ | *Base.* | $\phi_M$ |
| BAShapes | 80.50 | 72.02 | 82.67 | 47.50 | 73.69 | 100.00 | 80.50 | 72.81 | 83.00 | 47.50 | 82.98 | 100.00 |
| BBBP | 80.88 | 50.00 | 85.32 | 84.80 | 50.00 | 79.16 | 86.76 | 50.00 | 80.08 | 80.88 | 50.00 | 83.75 |
| Mutagenicity | 78.69 | 65.93 | 81.36 | 76.15 | 61.69 | 79.23 | 76.73 | 61.80 | 77.96 | 76.61 | 61.80 | 77.27 |
| IMDB | 74.50 | 35.35 | 74.16 | 73.50 | 25.67 | 76.00 | 74.00 | 25.67 | 75.00 | 76.00 | 26.98 | 72.50 |
| NCI1 | 70.56 | 60.04 | 73.81 | 70.07 | 59.25 | 69.70 | 70.56 | 61.84 | 74.20 | 70.32 | 57.67 | 68.11 |

# D  MORE DETAILS ON GROUNDING $\phi_M$ INTO THE INPUT FEATURE SPACE $\mathbf{X}$

## D.1  ON THE CANONICAL REPRESENTATION OF SUBGRAPH INPUT FEATURE $\mathbf{Z}$

To create a canonical representation, we partition the node set into orbits and establish a consistent ordering using Algorithm 1. The ordering scheme employs a hierarchical multi-criteria approach that ensures deterministic results:

1. **Anchor priority:** The orbit containing the anchor node is always placed first

2. **Size ordering:** Remaining orbits are sorted by cardinality in ascending order

3. **Degree signature:** Orbits with identical sizes are distinguished by their sorted degree sequences

4. **Distance profile:** Further ties are resolved using sorted distances from the anchor node

5. **Node identifiers:** Ultimate disambiguation is achieved through sorted node identifiers

We prove in Theorem D.1 that this multi-criteria lexicographic ordering produces a deterministic total order, ensuring the canonical representation can be reliably reproduced across identical graph structures.

---

**Algorithm 1:** Stable Orbit Decomposition with Anchor

---

**Input:** Graph $G$, anchor node $a$
**Output:** Orbit labels $L$ and sorted orbits $\mathcal{O}$
1 **Function** StableOrbitDecomposition$(G, a)$
2 $\quad$ $D \leftarrow$ *ComputeAnchorDistances*$(G, a)$ /* Calculate shortest path distances from anchor */
3 $\quad$ $\Sigma \leftarrow$ *FindAllAutomorphisms*$(G)$ /* Enumerate all graph symmetries */
4 $\quad$ $\mathcal{O}_{raw} \leftarrow$ *ExtractNodeOrbits*$(\Sigma, G)$ /* Group nodes into symmetry equivalence classes */
5 $\quad$ $\mathcal{O}_{anchor} \leftarrow$ *IdentifyAnchorOrbit*$(\mathcal{O}_{raw}, a)$ /* Locate orbit containing anchor node */
6 $\quad$ $\mathcal{O} \leftarrow$ *StableSortOrbits*$(\mathcal{O}_{raw}, \mathcal{O}_{anchor}, D, G)$ /* Sort by size, degree, distance, node IDs */
7 $\quad$ $L \leftarrow$ *AssignCanonicalLabels*$(\mathcal{O}, a)$ /* Map nodes to orbit labels with anchor priority */
8 $\quad$ **return** $(L, \mathcal{O})$

---

**Theorem D.1** (Orbit Sorting Consistency). *The multi-criteria orbit sorting scheme employed in Algorithm 1 produces a deterministic total ordering for any graph with fixed structure and anchor node.*

*Proof.* Let $\mathcal{O} = \{O_1, O_2, \ldots, O_k\}$ be the set of orbits obtained from the automorphism group decomposition. We define the sorting key for orbit $O_i$ as:

$$Key(O_i) = (|O_i|, \mathbf{d}_i, \mathbf{dist}_i, \mathbf{ids}_i) \tag{14}$$

where:

- $|O_i|$ is the orbit size

- $\mathbf{d}_i = \text{sorted}([\deg(v) : v \in O_i])$ is the sorted degree sequence

- $\mathbf{dist}_i = \text{sorted}([d_G(a, v) : v \in O_i])$ is the sorted distance sequence from anchor $a$

- $\mathbf{ids}_i = \text{sorted}(O_i)$ is the sorted node identifier sequence

We prove that this lexicographic ordering induces a strict total order on $\mathcal{O}$.

**Well-definedness:** Each component is well-defined for any finite graph: $|O_i| \in \mathbb{N}$, $\mathbf{d}_i \in \mathbb{N}^{|O_i|}$, $\mathbf{dist}_i \in (\mathbb{N} \cup \{\infty\})^{|O_i|}$, and $\mathbf{ids}_i$ is a finite sequence of distinct node identifiers.

**Totality:** For any two distinct orbits $O_i, O_j$ with $i \neq j$, we have $O_i \cap O_j = \emptyset$ by definition of orbit decomposition. We show $Key(O_i) \neq Key(O_j)$ by case analysis:

1. If $|O_i| \neq |O_j|$, then $Key(O_i) \neq Key(O_j)$ immediately.

2. If $|O_i| = |O_j|$ but $\mathbf{d}_i \neq \mathbf{d}_j$, then the orbits have different degree signatures, so $Key(O_i) \neq Key(O_j)$.

3. If $|O_i| = |O_j|$ and $\mathbf{d}_i = \mathbf{d}_j$ but $\mathbf{dist}_i \neq \mathbf{dist}_i$, then the orbits have different distance profiles from the anchor, so $Key(O_i) \neq Key(O_j)$.

4. If $|O_i| = |O_j|$, $\mathbf{d}_i = \mathbf{d}_j$, and $\mathbf{dist}_i = \mathbf{dist}_j$ but $\mathbf{ids}_i \neq \mathbf{ids}_j$, then the orbits contain different sets of nodes (since node identifiers are unique), so $Key(O_i) \neq Key(O_j)$.

**Tiebreaker completeness:** The final case cannot occur when $O_i = O_j$. Suppose $|O_i| = |O_j|$, $\mathbf{d}_i = \mathbf{d}_j$, $\mathbf{dist}_i = \mathbf{dist}_j$, and $\mathbf{ids}_i = \mathbf{ids}_j$. Then $\text{sorted}(O_i) = \text{sorted}(O_j)$. Since node identifiers are unique within a graph, this implies $O_i$ and $O_j$ contain exactly the same nodes, contradicting the assumption that $i \neq j$ (orbits are disjoint).

**Determinism:** Each component of $Key(O_i)$ is computed deterministically from the graph structure and anchor choice. Since lexicographic comparison admits no ties between distinct orbits and standard sorting algorithms are deterministic, the resulting orbit ordering is completely determined by the graph structure.

Therefore, the multi-criteria sorting scheme produces a unique, deterministic total ordering of orbits for any fixed graph structure and anchor node. □

*Remark* D.2. This result ensures that the stable orbit decomposition is reproducible across multiple runs for the same graph instance. The hierarchical sorting criteria are designed to resolve ties at progressively finer granularities, with the node identifier sequence providing ultimate disambiguation. Since our method operates on graphs with consistent structural representations, the deterministic ordering property is sufficient for ensuring algorithmic reliability.

### D.2 EXAMPLES WITH GUIDANCE ON READING GROUNDING RULE VISUALIZATIONS

We show an example of our generated explanations for a single predicate $p_{36}$ from the real-world dataset BBBP in Figure 9, with the grounding rule visualization presented in Figure 9(a). Note that each orbit in the visualization is labeled and colored differently. The figure illustrates three node orbits, such as Orbit 0 (the central red node) and Orbit 2 (the blue node), and an edge orbit, Orbit 3 (the four thick teal edges). This distinction improves the fine-grained expressiveness of our grounding rules, enhancing the structural specificity of the explanation.

The grounding rules are presented in Disjunctive Normal Form (DNF), where the overall explanation is a disjunction of cases (clauses) connected by an implicit "or." Each case is a conjunction of conditions on different orbits. To interpret these rules:

- **Case Structure**: In Figure 9(a), two cases are shown. Case 1 is the conjunction of the condition on Orbit 2 (`'#O = 1'`) and the condition on Orbit 3 (`'#(C, O) >= 1'`). Case 2 is the conjunction of the condition on Orbit 2 (`'#O = 0'`) and the conditions on Orbit 3 (`'#(C, O) >= 2 and #(C, N) = 0'`).

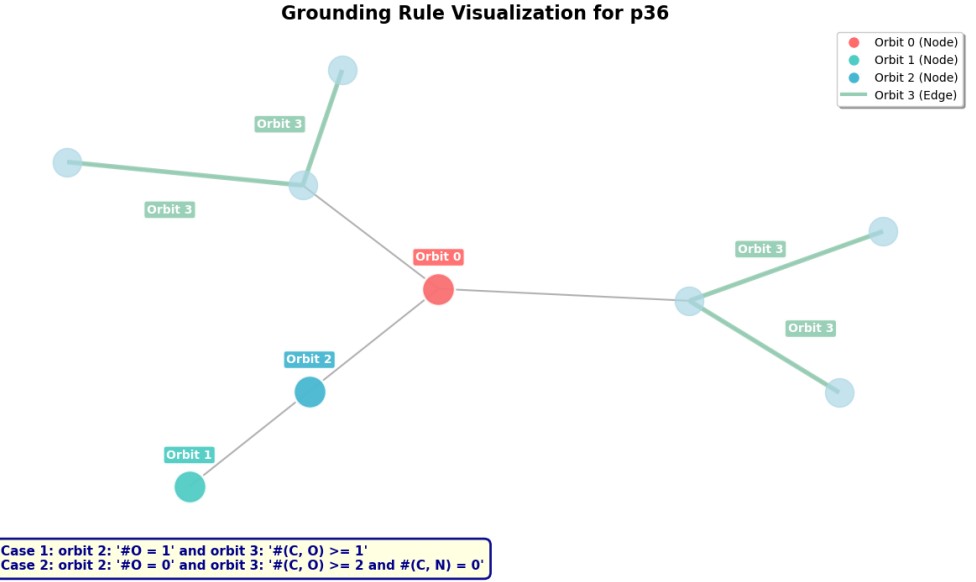

(a) An example of ground-rule explanations for a single predicate $p_{36}$.

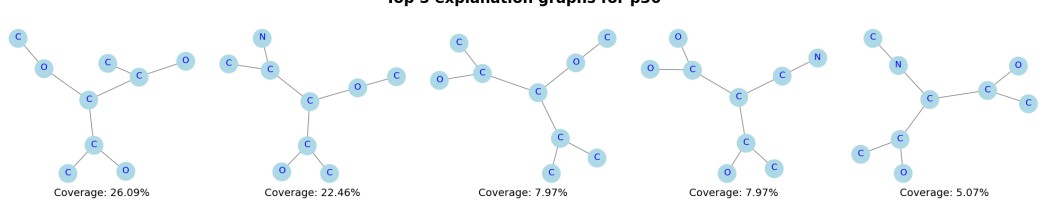

(b) Top 5 subgraph explanations for a single predicate $p_{36}$. Coverage below each subgraph indicates the percentage of instances that activate the predicate $p_{36}$ in which this subgraph occurs as an explanation.

Figure 9: An example of grounding rule explanations (top) and subgraph explanations (bottom) generated by our proposed approach $\phi_M$.

- **Node Orbit Interpretation**: Orbit 2 is a node orbit. The condition '#O = 1' in Case 1 means that the node(s) in Orbit 2 must include exactly one Oxygen (O) atom; '#O = 0' in Case 2 forbids Oxygen in that orbit.

- **Edge Orbit Interpretation**: Orbit 3 is an edge orbit. The condition '#(C, O) >= 2' in Case 2 means that at least two of the edges in Orbit 3 must connect a Carbon (C) atom to an Oxygen (O) atom, while '#(C, N) = 0' forbids Carbon–Nitrogen edges in that orbit.

- **Case Satisfaction**: A specific case is satisfied only if all of its conjunctive conditions are met simultaneously. The model's prediction is explained if the input graph satisfies at least one of the listed cases.

A major limitation of existing explanation methods is their *ineffective grounding*, as they often associate each concept with a single, possibly cherry-picked subgraph. Our approach provides a richer, data-driven alternative. As shown in Figure 9(b), we display the top-5 representative subgraphs for each rule, ranked by their frequency in the dataset. Coverage quantifies the proportion of instances activating predicate $p_{36}$ in which each subgraph occurs as an explanation, providing a more comprehensive view of the concept's presence across the data.

This strong alignment between our abstract rules and concrete examples is a direct result of our methodology. We first extract these subgraph instances directly from the data and then learn the general grounding rules from this empirical collection. The resulting dual representation—a formal logical rule paired with a ranked set of visual instances—offers a more detailed and multifaceted

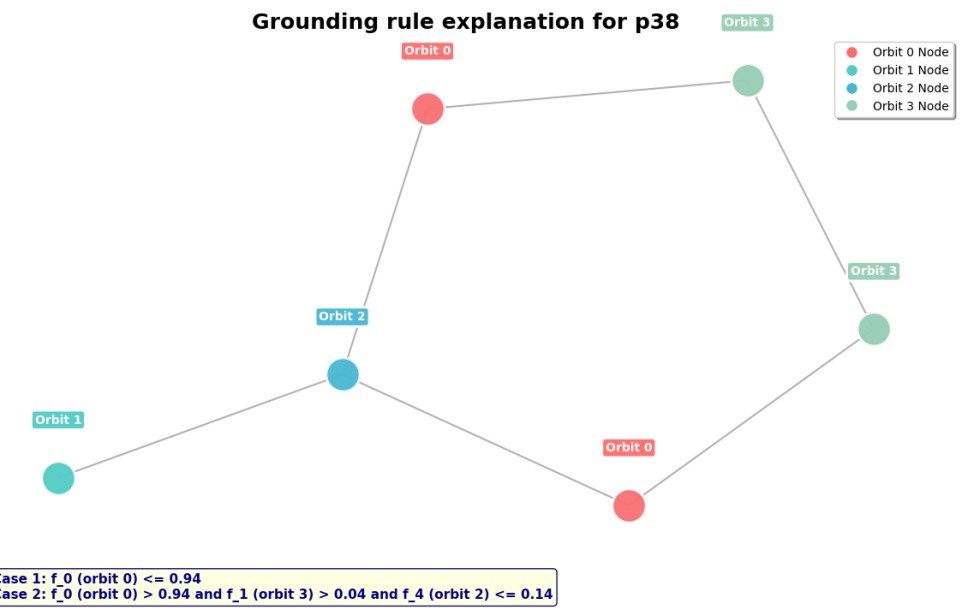

Figure 10: An example of grounding rule explanations on *continues* feature space $\mathbf{X}$ generated by our proposed approach $\phi_M$.

explanation than existing methods, yielding a deeper, more robust, and ultimately more interpretable grounding of the GNN's behavior.

### D.3    EXAMPLE: GROUNDING ($\phi_M$) IN A CONTINUOUS FEATURE SPACE

We demonstrate our method's ability to generate explanations in a continuous feature space using predicate $p_{38}$ as an example (Figure 10). For demonstration only, $p_{38}$ is trained on a synthetic dataset constructed by replacing discrete node features with continuous random features in a subset of Mutagenicity. To the best of our knowledge, most existing explanation approaches cannot handle such cases effectively. Their reliance on discrete attributes means their subgraph explanations lose meaning when faced with continuous feature distributions. Our approach addresses this limitation by leveraging orbit-based feature aggregation combined with learnable, threshold-based rules. The grounding rules are formulated in Disjunctive Normal Form (DNF), where different cases are connected with an implicit "or." To interpret these rules:

- **Case Structure**: Case 2 is a conjunction of three conditions: the condition on Orbit 0 (`f_0(orbit 0) > 0.94`), on Orbit 3 (`f_1(orbit 3) > 0.04`), and on Orbit 2 (`f_4(orbit 2) <= 0.14`).

- **Orbit-based Aggregation**: Each orbit aggregates continuous features from its constituent nodes using a statistical function (e.g., mean). This provides a robust summary of the features for all structurally equivalent nodes within the pattern.

- **Continuous Feature Interpretation**: The condition `f_0(orbit 0) > 0.94` means the aggregated value of the 0-th feature dimension across the nodes in Orbit 0 must exceed 0.94. Similarly, other conditions apply learned thresholds to different feature dimensions of their respective orbits.

- **Adaptive Threshold Learning**: These numerical thresholds (e.g., 0.94, 0.04, 0.14) are not fixed but are learned during training to define optimal boundaries that best discriminate between prediction classes.

- **Case Satisfaction**: Case 2 is satisfied only when all three of its threshold conditions are met simultaneously, ensuring that both structural and feature constraints work in conjunction.

Traditional subgraph-based explanations are ill-suited for this setting. They typically produce a single graph structure with discrete labels, which fails to capture the nuances of how continuous feature distributions influence a model's prediction. In contrast, our orbit-based rules provide a more robust and expressive explanation. By aggregating features across structurally equivalent nodes and learning discriminative thresholds, our method specifies *how* continuous feature values within a given topological pattern collectively drive the model's decision. This capability enables our method to deliver meaningful, interpretable explanations for complex real-world datasets where traditional approaches fail due to the prevalence of continuous attributes.

# E    ADDITIONAL DETAILS OF RULE-BASED EXPLANATIONS

## E.1    THE METRICS: COVERAGE, STABILITY, AND VALIDITY

To complement this qualitative analysis, we quantitatively evaluate the generated explanations using a set of objective and reproducible metrics that reflect their practical utility to end users:

- **Coverage:** The proportion of target-class instances for which the rule-based prediction remains correct *when restricted to only valid subgraph patterns* (i.e., after removing all invalid patterns). Formally,

$$\text{Coverage}(\phi) = \frac{|\{x \in \mathcal{D}_c : \phi(x) = \text{True}\}|}{|\mathcal{D}_c|}, \tag{15}$$

  where $\phi(x)$ is True iff evaluating the rule on $x$ using only valid subgraph patterns (i.e., considering their presence or absence) predicts class $c$. Here, "valid" refers to subgraph patterns that appear as actual subgraphs of molecules in the dataset, ensuring they are derived from structurally valid molecular graphs rather than artificially constructed or chemically impossible fragments.

- **Stability:** The consistency of explanation subgraphs across multiple runs with different random seeds. This metric is crucial for building user trust, as inconsistent explanations undermine confidence in the model's reasoning. We measure stability as the fraction of subgraphs repeated in *all runs* relative to the largest number of subgraphs in any single run:

$$\text{Stability}(\{\phi_i\}_{i=1}^{k}) = \frac{|\bigcap_{i=1}^{k} \phi_i|}{\max_{i=1,\ldots,k} |\phi_i|}, \tag{16}$$

  where $\phi_i$ denotes the set of subgraphs discovered in the $i$-th run and $k$ is the total number of runs. In this case of $k = 3$ (3 runs), this simplifies to

$$\text{Stability} = \frac{|\phi_1 \cap \phi_2 \cap \phi_3|}{\max\{|\phi_1|, |\phi_2|, |\phi_3|\}}. \tag{17}$$

- **Validity:** The proportion of explanation subgraphs that correspond to valid chemical fragments or structural motifs found in the dataset. For molecular datasets, this ensures that the generated explanations respect chemical constraints and represent realistic molecular substructures. Invalid fragments (e.g., impossible bond configurations or non-existent functional groups) reduce the practical utility of explanations for domain experts. We define validity as:

$$\text{Validity} = \frac{|\{f \in \Phi : f \in \mathcal{F}_{\text{valid}}\}|}{|\Phi|}, \tag{18}$$

  where $\Phi$ is the set of all generated fragments and $\mathcal{F}_{\text{valid}}$ represents the set of chemically valid fragments derived from the training data.

In Figures 4 and 5, we normalized the complexity of the generated rule-based explanations for all approaches for clearer visualization. Specifically, we configured each approach—setting the tree depth for $\phi_M$, specifying the number of concepts for GRAPHTRAIL, and selecting the top-$k$ subgraphs for GLGEXPLAINER—to generate rules of a comparable scale, using 3 concepts (or predicates) per class. This choice does not affect the fundamental nature of the methods, and we maintain hyperparameter and random seed settings consistent with the results in Table **??**.

For the stability experiment in Section 4.2, however, a different complexity was required. A fair comparison of stability necessitates normalizing explanation complexity to a level that is both challenging and informative. Through preliminary analysis, we found that a low complexity setting (e.g., 3 concepts) was insufficiently discriminative for a rigorous comparison. In this setting, our method achieved near-perfect stability, creating a ceiling effect that, while demonstrating its robustness, prevented a more nuanced assessment of relative performance against the baselines. To create a more challenging benchmark that allows for a fine-grained evaluation across all methods, we chose a moderate complexity of approximately 5–6 concepts per class, as this was empirically determined to be the most informative for comparing the stability of the different approaches. For all other metrics, we use the default hyperparameter and seed settings for each explainer, consistent with Table **??**.

## E.2 FURTHER ANALYSIS OF TABLES 3 AND 4

Table 3 reveals an important distinction between the baselines in the binary classification setting. GLGEXPLAINER produces rules for *both* classes, but these rules are often conflicting and rely on less representative subgraphs, resulting in very low coverage (e.g., only 6.11% for Mutagenicity class 0). Another major limitation of GLGEXPLAINER is its reliance on prior knowledge and its high sensitivity to hyperparameter choices—an issue explicitly acknowledged by the original authors (Azzolin et al., 2023). This makes the method less applicable to new datasets; for example, despite replicating their settings with direct guidance from the authors, GLGEXPLAINER failed to learn any rules on BBBP. We discuss these reproducibility challenges in more detail in Section B.3. By contrast, GRAPHTRAIL generates only *unilateral* rules. This design makes it artificially easier for GRAPHTRAIL to appear correct—since all instances of the opposite class are explained by default—but at the cost of explanatory quality, as the method produces only discriminative rules rather than descriptive rules for each class. Furthermore, because GRAPHTRAIL frequently produces chemically invalid subgraphs, many of its explanations cannot be considered constructive, as they depend merely on the presence or absence of subgraphs that never occur in the dataset, further reducing effective coverage. We also consulted with the original authors and confirmed this issue, as discussed in Section B.3. This combination of unilateral shortcuts and invalid fragments underscores why GRAPHTRAIL's explanations are unsuitable for faithful or practical interpretability.

In sharp contrast, our approach $\phi_M$ learns rules for *both* classes while still achieving very high coverage (e.g., 80.06% and 82.58% for Mutagenicity, and 47.86% and 98.16% for BBBP). Achieving strong performance under this stricter and more balanced setting is particularly noteworthy, as it demonstrates that $\phi_M$ provides class-wise explanations without relying on unilateral shortcuts or invalid patterns. This fairness in rule construction makes the comparison against baselines more rigorous and highlights the strength of our method in generating meaningful explanations that remain faithful to the underlying model.

Table 4 complements this picture with *stability* and *validity*. $\phi_M$ attains the highest stability across seeds (66.67% on Mutagenicity, 60.00% on BBBP), surpassing GLGEXPLAINER on Mutagenicity (40.00%) and GRAPHTRAIL (37.50% on Mutagenicity, 20.00% on BBBP). High stability suggests that $\phi_M$ learns rules that are robust to randomness in training and sampling. On *validity*, $\phi_M$ reaches $100.00\% \pm 0.00$ on both datasets, matching GLGEXPLAINER on Mutagenicity but far exceeding GRAPHTRAIL ($61.90\% \pm 6.73$ on Mutagenicity and 0.00% on BBBP).

Taken together, the results across coverage, stability, and validity form a consistent narrative: while GLGEXPLAINER suffers from conflicting rules and poorly representative explanations, and GRAPH-TRAIL exploits unilateral shortcuts compounded by invalid and equally unrepresentative subgraphs, $\phi_M$ generates balanced rules for both classes that are faithful, reproducible, and chemically valid. This balance is particularly rare in explanation methods, underscoring the robustness and scientific reliability of $\phi_M$ as a framework for generating high-quality graph explanations.

**Why don't we use a human study in this work to assess the final explanations?** While human studies are widely employed in XAI for interpretability assessment, they are particularly ill-suited for scientific domains like biochemistry for several key reasons: (1) Meaningful evaluation in such contexts demands *domain expertise*, making large-scale recruitment of qualified participants prohibitively difficult and expensive. (2) Laypeople's subjective perceptions of "understandability" often diverge significantly from *scientific validity*—explanations that appear intuitively clear may be biochemically erroneous or fundamentally misleading. (3) Human evaluations inherently introduce

substantial variability due to differences in participant expertise levels, experimental design choices, and evaluation criteria, thereby compromising the reproducibility essential for scientific validation.

We therefore adopt objective, quantitative metrics, *coverage*, *stability*, and *validity*, specifically designed to evaluate biochemical explanation quality. This framework prioritizes generalizability, consistency, and scientific accuracy, providing a rigorous alternative to subjective assessments that aligns with the precision requirements of scientific inquiry.

### E.3  MORE EXAMPLES

We present the generated rule-based explanations of our approach $\phi_M$ on BAShapes, BBBP, Mutagenicity, and IMDB. NCI1 is excluded because the information on its feature attributes is not available. Although the features are one-hot encoded for atoms, the mapping between atoms and feature indices has not been released, so we chose not to report results for NCI1.

**IMDB Dataset Classification Rules** (Depth = 10)

$(\neg p_0 \wedge \neg p_{274} \wedge \neg p_{267} \wedge \neg p_{268} \wedge \neg p_{16} \wedge \neg p_{270} \wedge \neg p_{276} \wedge \neg p_{333} \wedge \neg p_{278} \wedge \neg p_{282})$
$\vee\, (\neg p_0 \wedge \neg p_{274} \wedge \neg p_{267} \wedge \neg p_{268} \wedge p_{16})$
$\vee\, (p_0) \Rightarrow$ IMDB Class 0;

$(\neg p_0 \wedge \neg p_{274} \wedge \neg p_{267} \wedge \neg p_{268} \wedge \neg p_{16} \wedge \neg p_{270} \wedge \neg p_{276} \wedge \neg p_{333} \wedge \neg p_{278} \wedge p_{282})$
$\vee\, (\neg p_0 \wedge \neg p_{274} \wedge \neg p_{267} \wedge \neg p_{268} \wedge \neg p_{16} \wedge \neg p_{270} \wedge \neg p_{276} \wedge \neg p_{333} \wedge p_{278})$
$\vee\, (\neg p_0 \wedge \neg p_{274} \wedge \neg p_{267} \wedge \neg p_{268} \wedge \neg p_{16} \wedge \neg p_{270} \wedge \neg p_{276} \wedge p_{333})$
$\vee\, (\neg p_0 \wedge \neg p_{274} \wedge \neg p_{267} \wedge \neg p_{268} \wedge \neg p_{16} \wedge \neg p_{270} \wedge p_{276})$
$\vee\, (\neg p_0 \wedge \neg p_{274} \wedge \neg p_{267} \wedge \neg p_{268} \wedge \neg p_{16} \wedge p_{270})$
$\vee\, (\neg p_0 \wedge \neg p_{274} \wedge \neg p_{267} \wedge \neg p_{268} \wedge p_{268})$
$\vee\, (\neg p_0 \wedge \neg p_{274} \wedge \neg p_{267} \wedge p_{267})$
$\vee\, (\neg p_0 \wedge p_{274}) \Rightarrow$ IMDB Class 1

**BAShapes Class 0 Rules:** (Depth = 5)

$(\neg p_{280} \wedge \neg p_{52} \wedge \neg p_{324} \wedge \neg p_{55} \wedge \neg p_{4495})$
$\vee\, (\neg p_{280} \wedge \neg p_{52} \wedge \neg p_{324} \wedge p_{55} \wedge \neg p_{14})$
$\vee\, (\neg p_{280} \wedge \neg p_{52} \wedge p_{324} \wedge \neg p_{587} \wedge p_{49})$
$\vee\, (\neg p_{280} \wedge \neg p_{52} \wedge p_{324} \wedge p_{587})$
$\vee\, (\neg p_{280} \wedge p_{52} \wedge \neg p_8 \wedge \neg p_{324} \wedge \neg p_{55})$
$\vee\, (\neg p_{280} \wedge p_{52} \wedge p_8 \wedge \neg p_{36} \wedge p_{1207})$
$\vee\, (\neg p_{280} \wedge p_{52} \wedge p_8 \wedge p_{36})$
$\vee\, (p_{280} \wedge \neg p_{204} \wedge \neg p_{698} \wedge p_{1817})$
$\vee\, (p_{280} \wedge \neg p_{204} \wedge p_{698})$
$\vee\, (p_{280} \wedge p_{204})$
$\Rightarrow$ BAShapes Class 0

**BAShapes Class 1 Rules:** (Depth = 5)

$(\neg p_{280} \wedge \neg p_{52} \wedge \neg p_{324} \wedge \neg p_{55} \wedge p_{4495})$
$\vee\, (\neg p_{280} \wedge \neg p_{52} \wedge \neg p_{324} \wedge p_{55} \wedge p_{14})$
$\vee\, (\neg p_{280} \wedge \neg p_{52} \wedge p_{324} \wedge \neg p_{587} \wedge \neg p_{49})$
$\vee\, (\neg p_{280} \wedge p_{52} \wedge \neg p_8 \wedge \neg p_{324} \wedge p_{55})$
$\vee\, (\neg p_{280} \wedge p_{52} \wedge \neg p_8 \wedge p_{324})$
$\vee\, (\neg p_{280} \wedge p_{52} \wedge p_8 \wedge \neg p_{36} \wedge \neg p_{1207})$
$\vee\, (p_{280} \wedge \neg p_{204} \wedge \neg p_{698} \wedge \neg p_{1817} \wedge \neg p_{113})$
$\vee\, (p_{280} \wedge \neg p_{204} \wedge \neg p_{698} \wedge \neg p_{1817} \wedge p_{113})$
$\Rightarrow$ BAShapes Class 1

**BBBP Dataset Classification Rules** (Depth = 3)

$(\neg p_{38} \wedge \neg p_5 \wedge p_{31}) \vee (\neg p_{38} \wedge p_5 \wedge \neg p_{59}) \vee (\neg p_{38} \wedge p_5 \wedge p_{59}) \vee (p_{38} \wedge \neg p_{61} \wedge p_{20})$
$\vee\, (p_{38} \wedge p_{61} \wedge \neg p_{54}) \vee (p_{38} \wedge p_{61} \wedge p_{54}) \Rightarrow$ BBBP Class 0;

$(\neg p_{38} \wedge \neg p_5 \wedge \neg p_{31}) \vee (p_{38} \wedge \neg p_{61} \wedge \neg p_{20}) \Rightarrow$ BBBP Class 1

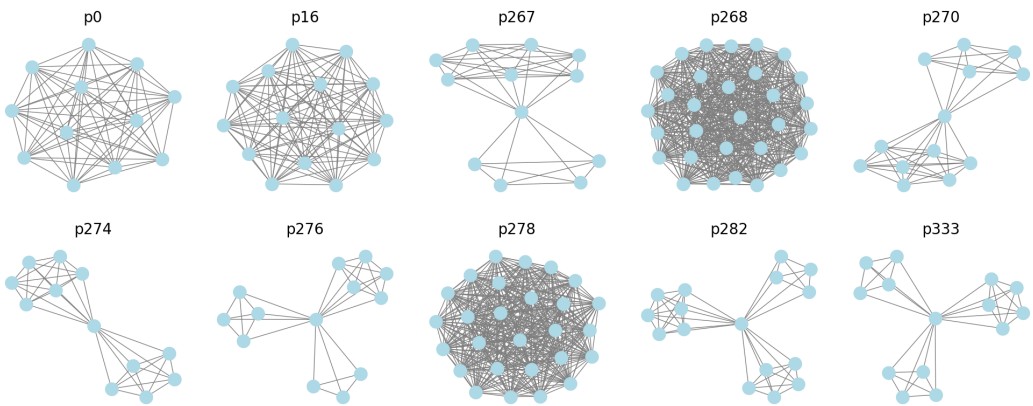

Figure 11: Our approach's grounded explanation ($\phi_M$) for IMDB.

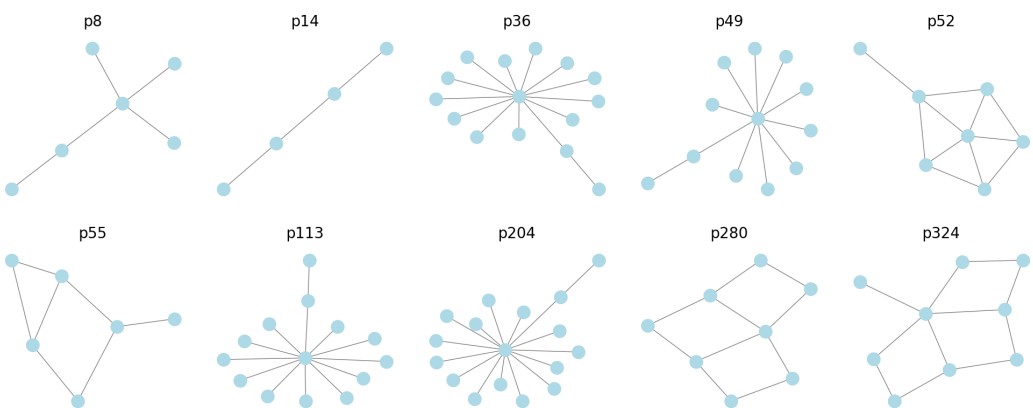

Figure 12: Our approach's grounded explanation ($\phi_M$) for BAMultiShapes.

**Mutagenicity Dataset Classification Rules** (Depth = 3)

$(\neg p_2 \wedge p_{20} \wedge \neg p_{66}) \vee (p_2 \wedge \neg p_{38} \wedge \neg p_{64}) \vee (p_2 \wedge p_{38} \wedge p_{17}) \Rightarrow$ Mutagenicity Class 0;

$(\neg p_2 \wedge \neg p_{20} \wedge \neg p_1) \vee (\neg p_2 \wedge \neg p_{20} \wedge p_1) \vee (\neg p_2 \wedge p_{20} \wedge p_{66}) \vee (p_2 \wedge \neg p_{38} \wedge p_{64})$
$\vee (p_2 \wedge p_{38} \wedge \neg p_{17}) \Rightarrow$ Mutagenicity Class 1

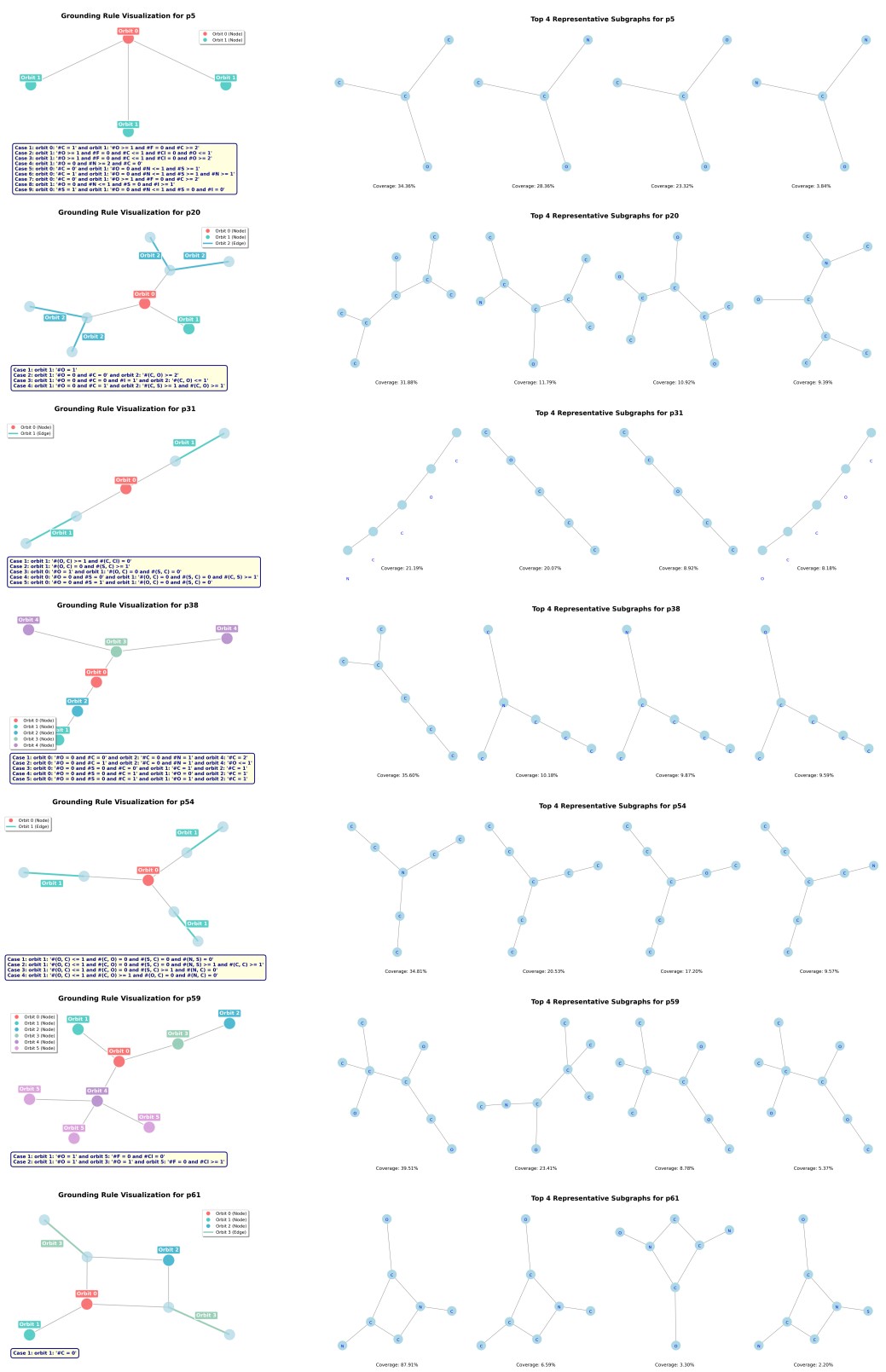

Figure 13: Our approach's grounded explanation ($\phi_M$) for BBBP.

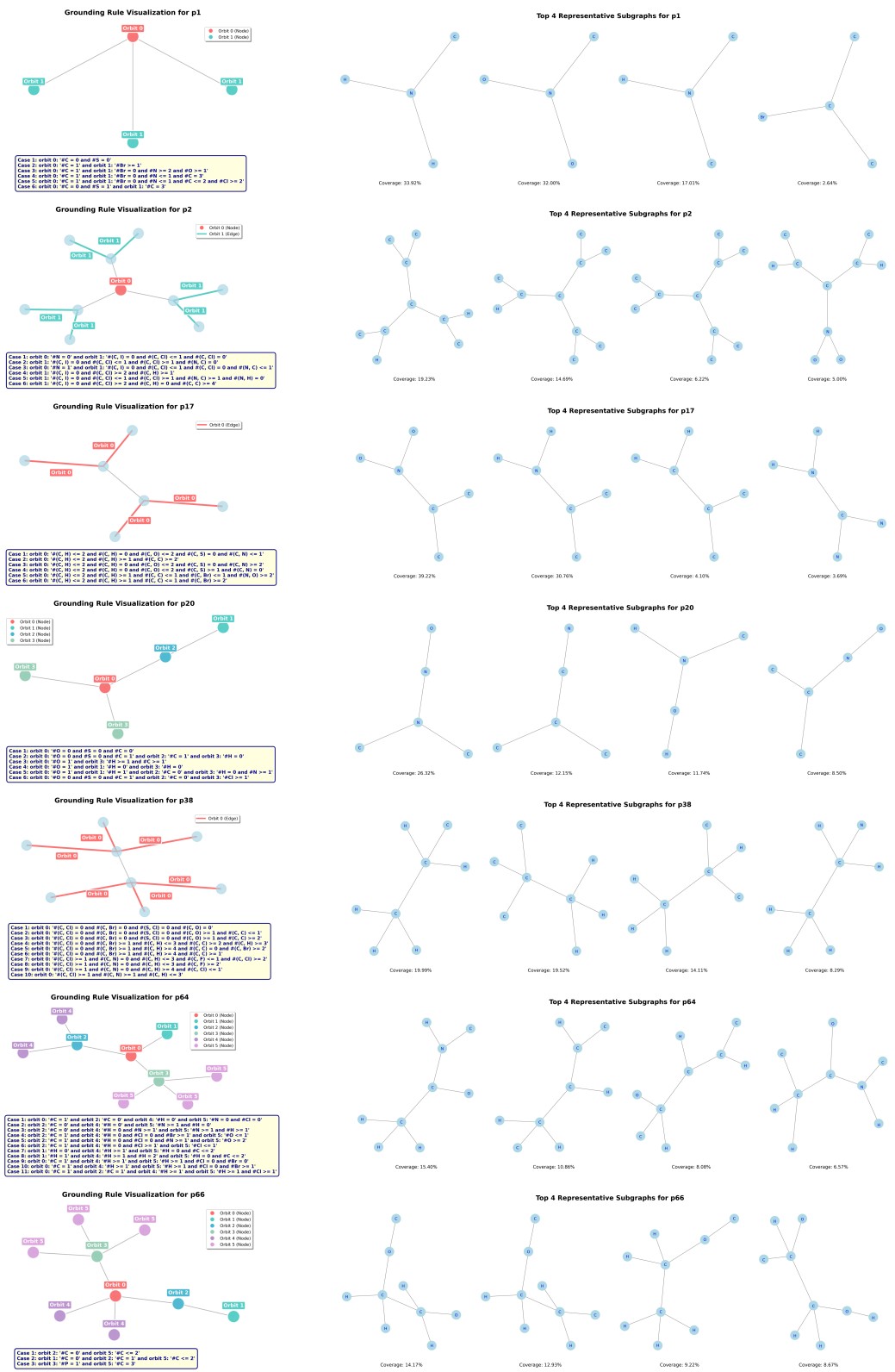

Figure 14: Our approach's grounded explanation ($\phi_M$) for Mutagenicity.

## F  LLM Usage

Large Language Models (LLMs) were used as a general-purpose assistive tool in the preparation of this work. Specifically, LLMs supported tasks such as refining the clarity of writing, suggesting alternative phrasings, and checking the consistency of technical terminology. They were **not** used for generating research ideas, conducting experiments, or producing original scientific contributions. All substantive research decisions, analysis, and results presented in this paper are the responsibility of the authors. The authors have carefully reviewed and verified all LLM-assisted text to ensure accuracy and originality.

