# OpenReview forum: "LogicXGNN:  Grounded Logical Rules for Explaining Graph Neural Networks"
_ICLR.cc/2026/Conference — ICLR 2026 Poster_

### Official Review · Reviewer_kSni · 2025-11-02

**Soundness:** 1
**Presentation:** 1
**Contribution:** 3
**Rating:** 4
**Confidence:** 4

**Summary:**

This paper tackles post-hoc rule-based model-level GNN explainabiltiy, aiming to ground the extracted global rules to human understandable data-level. They propose LOGICXGNN, which constructs logical rules directly grounded in data, ensuring that explanations correspond to real subgraphs. It introduces a new metric, data-grounded fidelity (FidD), which measures how well explanations match the GNN’s outputs in the actual input space, not in an abstract one.

**Strengths:**

1. Quantitative results are consistently strong. Strong empirical performance on fidelity, coverage, stability, validity and efficiency.
2. Well motivated. The paper identifies a valid and underexplored problem. Clear identification of an overlooked issue in prior rule-based explainers.
3. The framework is complete and reproducible.

**Weaknesses:**

1. **Missing clear discussion and comparison on an important baseline.** The paper briefly mentions GCNeuron (Xuanyuan et al., AAAI 2023) in the related work section, categorizing it as a “concept-based global explanation”. However, it does not clearly articulate the methodological differences or justify the absence of a direct comparison. Given that GCNeuron is also a rule-based, model-level GNN explainer that produces logical rules to characterize model behavior, a clearer discussion of methodological distinctions, or at least a brief justification for not comparing, would strengthen the completeness of the evaluation.
2. **Overly domain-specific evaluation. Lack of validation on simple synthetic tasks.**
The paper claims to propose a general and scalable global rule-based GNN explainer, but almost all examples and visualizations are from molecular datasets (Mutagenicity, BBBP, NCI1). This makes the work look domain-specific rather than general. The authors did not include enough simple synthetic datasets where the ground-truth reasoning patterns are known. On such datasets, an effective rule-based method should ideally achieve near 100% data-grounded fidelity, clearly showing that it can capture the model’s true decision rules. Without this type of controlled experiment, it remains unclear whether the proposed rules genuinely reflect model reasoning or are just fitting chemical regularities in the data.
3. **Excessive complexity of grounded rules without well-clarifications.** While the proposed orbit-based grounding mechanism improves formal precision, it significantly increases the structural and logical complexity of the resulting rules. The grounded rules involve multi-level orbit decomposition, nested logical conjunctions, and numerical thresholds, which make them difficult for non-expert users to interpret. Compared to prior rule-based explainers such as GLGExplainer and GraphTrail that generate concise and human-readable logic, the grounded rules here are overly abstract and cumbersome. This undermines one of the central goals of grounding in this paper, which was explicitly framed as aiming for "human-understandable" explanations.
4. **Insufficient justification for the rule-based paradigm.** Although the paper emphasizes the importance of rule grounding, it does not convincingly demonstrate why a rule-based approach is preferable to other established global explanation paradigms, such as generation-based (XGNN, GNNInterpreter) or subgraph-based (TreeX) methods. There is no human study, no qualitative comparison of interpretability, and no synthetic benchmark explicitly designed to evaluate rule quality. As a result, the claimed advantage of rule-based grounding remains conceptually appealing but empirically unproven.
5. **Questionable novelty of the proposed evaluation metric.** The proposed “Data-grounded Fidelity (Fid$^D$)” metric appears conceptually aligned with the instance-level fidelity evaluation already used in prior work such as TreeX, where global explanations are mapped back to instances to check whether they reproduce the original GNN’s predictions. The paper reframes this idea under a new term “data-grounded fidelity”, but this seems more like a terminological reframing than a fundamentally new metric design. Moreover, TreeX also provides motif-level global explanations evaluated on instances, making it a highly relevant baseline. However, the authors neither compare with TreeX nor acknowledge this conceptual overlap. This omission weakens the claimed novelty of Fid$^D$ and leaves the evaluation incomplete.
6. **Limited technical novelty.** The proposed framework largely combines existing ideas: constructing logical rules from learned predicates (as in GLGExplainer, GCNeuron and GraphTrail), grounding them to data (as in motif-based explainers like TreeX and PAGE), and evaluating instance-level fidelity (similar to TreeX).

**Questions:**

n/a

---

> ### Author Response · Authors · 2025-11-13
> **Initial Rebuttal to the Review kSni (Weaknesses 1–4)**
>
> We thank the reviewer for thoughtful comments and clarify several potential misunderstandings below.
>
> ---
> ### W1: Missing clear discussion and comparison on an important baseline.
>
> *Response:*  We would like to clarify that GCNeuron is a *concept-based* global explanation method. It analyzes which base concepts are detected by **individual neurons** in a GNN and measures their relative importance. However, **GCNeuron does not synthesize these neuron-level concepts into explicit class-wise logical rules that summarize the GNN’s overall decision-making process** [1]. While one could select neuron-level concepts associated with a particular class, they are not further combined into a single, coherent class-wise rule, as is done in our work.
>
> The recent work GraphTrail also explicitly excludes GCNeuron from its baselines, stating that GLGExplainer is the only comparable rule-based GNN explainer. Following the same rationale, we did not include it as a baseline. We thank the reviewer for this suggestion and will clarify this in the final revision.
>
> ---
> ### W2: Overly domain-specific evaluation. Lack of validation on simple synthetic tasks.
>
> *Response:*  We agree that validation on simple synthetic tasks is important for confirming that the extracted rules genuinely reflect model reasoning. In fact, **we performed several controlled experiments on synthetic tasks during development to validate our approach, two of which are already included in the paper** despite space constraints.
>
> *Case 1* is a simple GNN classification task (Fig. 2), where graphs 1–3 lack degree-2 nodes while graphs 4–5 contain them. The trained GNN achieved 100 % accuracy, and our method achieved 100 % fidelity, with the extracted rule (Eq. 11) exactly matching the ground truth.
>
> *Case 2* uses the BAMultiShapes dataset (Tab. 1), a more complex synthetic benchmark of 1,000 graphs formed by attaching House, Wheel, and Grid motifs to random Barabási–Albert graphs. Our method achieves 100% fidelity for both GraphSAGE and GAT (Tab. 13) and correctly identifies the underlying motif rules (Appendix E.3).
>
> We will clarify these synthetic task validations more explicitly in the revision.
>
> ---
> ### W3: Excessive complexity of grounded rules without well-clarifications.
>
> *Response:* We would like to clarify that our framework is intentionally designed to support *two complementary explanation modes*:
>
> - **Representative subgraphs** that match the complexity and style of prior methods (see the first row of Fig. 5), and
> - **Fully grounded logical rules** (the second row of Fig. 5) that provide formal precision when desired.
>
> In other words, **our method can output explanations at the same simplicity level as existing work**, while additionally offering a more expressive, formally grounded option for users who require it. Notably, our rules are presented in **DNF form**, which is more structured and interpretable than the formats used in prior methods.
>
> Overall, the framework offers a **spectrum of interpretability**, allowing users to choose the level of detail they prefer—from concise, human-readable subgraphs to fully grounded, formally precise rules. We will make this trade-off and its benefits clearer in the final revision.
>
> ---
> ### W4: Insufficient justification for the rule-based paradigm.
>
> *Response:*  We appreciate the reviewer’s concern and would like to clarify the intended scope of our work. Our paper builds on a line of prior studies (e.g., GCNeuron, GLGExplainer) that **have already compared rule-based explanations with other global paradigms**, including generation-based approaches such as XGNN. These works have shown that rule-based frameworks often offer clearer interpretability, stronger alignment with human concepts, and better contextual grounding. For example, GCNeuron demonstrates how rule-based analysis can organize the behaviors of individual neurons into distinct, human-intuitive concepts with explicit logical descriptions, whereas XGNN’s generated graphs often lack clear semantics, may fall outside the data distribution, and rely on additional black-box RL agents for explanation [2].
>
> Given that this paradigm-level comparison has been explored in prior literature, our work focuses on a different gap: **improving the faithfulness and formal grounding of rule-based explanations themselves**. Our contribution lies in showing that logical rule explanations can be made both *causally faithful* and *formally precise*, addressing limitations that earlier rule-based methods left open.
>
> Consistent with recent rule-based explainers such as GraphTrail, we therefore treat paradigm-level comparisons as established background rather than part of the core scope of our contribution.
>
> ---
> ### Reference:
> [1] Armgaan et al., GraphTrail: Translating GNN Predictions into Human-Interpretable Logical Rules, NeurIPS 2024.
>
> [2] Xuanyuan et al., Global Concept-Based Interpretability for Graph Neural Networks via Neuron Analysis, AAAI 2023.

---

> ### Author Response · Authors · 2025-11-13
> **Initial Rebuttal to the Review kSni (Weaknesses 5–6)**
>
> Following our earlier rebuttal addressing Weaknesses 1–4, we now clarify potential misunderstandings related to Weaknesses 5 and 6.
>
> ---
>
> ### W5: Questionable novelty of the proposed evaluation metric.
>
> *Response:* We acknowledge that both TreeX [3] and our proposed fidelity metric share a high-level goal of evaluating fidelity directly using subgraphs. However, despite this conceptual similarity, they operate in *fundamentally different computational spaces*. TreeX employs a **score-based aggregation** mechanism to explain individual instances—its fidelity is computed by summing weighted scores to determine which class best fits an instance. This represents a different explanation style from *logic rule–based approaches* such as GLGExplainer, GraphTrail, and our method, which rely on predicate evaluation and logic rule matching rather than numerical score aggregation. Consequently, our *Data-grounded Fidelity* resides in the **symbolic logic space**, directly assessing rule satisfaction rather than score accumulation—analogous to the difference between *analog* and *digital* systems.
>
> Moreover, according to the TreeX paper itself [3]:
> > “Note that fidelity is a metric calculated on individual instances, so we cannot compute fidelity for existing global-level methods, as they are unable to explain local instances like TreeX.”
>
> This statement from the original authors further underscores the computational and conceptual differences between the two fidelity formulations. Specifically, TreeX’s fidelity cannot be applied to global logic-rule methods, whereas our metric is explicitly designed to evaluate such global explanations. As such, TreeX explicitly excludes global-level (model-level) methods from quantitative comparison. We therefore consider it *conceptually relevant but not a suitable direct baseline* for our study. Accordingly, we will discuss it in the related work section in the revision, focusing on the high-level similarity in evaluation intent.
>
>
> ---
>
> ### W6: Limited technical novelty.
>
> *Response:* We appreciate the reviewer’s perspective and would like to clarify the source of potential misunderstanding. The comment seems to refer to high-level characteristics shared by an entire line of rule-based explanation research (e.g., constructing logical rules from learned predicates), rather than the specific mechanisms introduced in our framework. These broad descriptions capture the general paradigm, not the concrete technical innovations that distinguish one method from another.
>
>
> Under such a broad characterization, many follow-up advances in a research direction could appear “non-novel,” since they naturally inherit the same paradigm-level structure. Our contribution instead lies in **several distinct technical innovations** beyond prior logic rule-based explainers:
>
> 1. **Grounding quality and formal faithfulness:** We identify and address the previously overlooked issue of *grounding correctness* in rule-based explanations. Our framework explicitly ensures that extracted rules are formally linked to model reasoning through a structurally validated grounding process.
> 2. **Decoupling structural patterns from embedding patterns for concept learning:** Unlike prior logic rule-based methods that entangle embedding and structural patterns during concept learning, our approach explicitly disentangles the WL-hashing-based structural patterns, where message passing occurs, from the embedding dynamics. This separation mitigates the information loss that typically arises during message propagation and enables our method to more faithfully recover the true decision-making process of GNNs. *This contribution has been positively recognized by Reviewers nxGX and AscN.*
> 3. **Lightweight and efficient design:** Our method employs only simple, fast-to-train components (e.g., decision trees), avoiding the costly local explainer pipelines relied upon by prior work. This results in strong fidelity while significantly improving runtime efficiency.
> 4. **Engineering design and formalization of rule grounding:** We introduce several principled design choices that enhance the determinism, consistency, and scalability of rule grounding across diverse graph domains. We also retain subgraphs as an alternative explanation to give users greater flexibility.
>
>
> Together, these innovations form a technically distinct and practically valuable framework that advances the state of logic rule-based GNN explanation.
>
> ---
>
> ### Reference:
>
> [3] Shengyao et al., TreeX: Generating Global Graphical GNN Explanations via Critical Subtree Extraction, arXiv preprint, 2025.
>
> ---
>
> ### Final Remarks
>
> We hope our responses have addressed the reviewer’s concerns and clarified any potential misunderstandings. We respectfully ask the reviewer to kindly reconsider their score in light of these clarifications.

---

> ### Author Response · Authors · 2025-11-28
>
> Hello, thank you again for your detailed review. Since the rebuttal period is coming to an end, we wanted to note that we have carefully addressed your comments above. If any part remains unclear, we would be happy to clarify.

---

> > ### Comment · Reviewer_kSni · 2025-11-28
> >
> > We thank the author for the rebuttal, and thank you for putting efforts to address the concerns. However, the current responses do not fully resolve the concerns raised (especially W4, W5). From my own perspective, this paper itself is a borderline paper, which I would give a 5. The novelty and contribution is not strong but acceptable, as long as the authors could address the small concerns carefully. We were waiting for the authors to update the manuscript as they explicitly promised in the rebuttal. It is now near the end of the discussion phase, and no revised version has been provided. Therefore, I will keep the score for now.
> >
> > W1: I explicitly mentioned GCNeuron not because of whether it is concept-based or rule-based, but because it is also a model-level global explainer, which is exactly the scope of this paper. Labeling it as “concept-based” does not change this fact. It also provides class-level concepts and combinational logic. Referencing GraphTrail’s omission does not justify omitting it here. Each paper needs to clarify its own methodological boundaries. A brief explanation of how LogicXGNN differs from GCNeuron at the model level, and why it is not considered as a baseline, would make the positioning more complete.
> >
> > W2: The provided “synthetic” cases (a trivial degree-based toy example and BAShapes) do not address my concern, because neither of them is a controlled rule-based synthetic benchmark with ground-truth logical rules. They test motif detection, not rule learning. Therefore, the core concern remains unaddressed.
> >
> > W3: The distinction between “representative subgraphs” and “fully grounded rules” does not address my concern. The core issue is that the grounded rules themselves remain overly complex and difficult to interpret, and providing an optional simpler view does not solve the fact that the main proposed contribution, while the grounded rules in this paper is not human-understandable in practice. (This W3 is an acceptable concern, and I won’t go too much into it. But since ICLR is a leading venue, the community may follow whatever explanation form gets accepted, and adopting an overly complex and not human-understandable rule format may not be a good idea.)
> >
> > W4: Prior work comparing rule-based and generation-based explainers does not remove the need for this paper to justify its own paradigm choice. My concern is about the positioning of this work, and the rebuttal does not explain why rule-based grounding is necessary or preferable, so the core justification remains unaddressed.
> >
> > W5&W6: My point was that the proposed Fid$^D$ is conceptually aligned with existing instance-level fidelity used in prior work, and the explanation mechanism (symbolic vs score-based) is not the metric itself. The response shifts to representation differences rather than explaining the actual novelty of the metric, so the concern remains unaddressed. The core pipeline largely follows prior rule-based explainers, and the rebuttal does not demonstrate substantial algorithmic or conceptual innovation beyond this paradigm. Therefore, the concern about limited technical novelty still stands.
> >
> > Given the unresolved concerns summarized above, I personally do not lean toward acceptance of this paper. That said, I acknowledge the differing views of the other reviewers, and I am fine with the final decision the AC considers appropriate.

---

> ### Author Response · Authors · 2025-11-28
>
> Thank you for your follow-up. We would like to note that uploading a revised PDF during the rebuttal period is **optional** under ICLR policy. We therefore focused on providing detailed, point-by-point responses in the discussion thread. Most concerns are relatively minor and primarily involve clarification rather than substantial changes. Accordingly, when we refer to the *revision* or *final revision*, we mean that—if accepted—we will incorporate these clarifications in the camera-ready version, **not as a revised manuscript during the rebuttal period.**
>
> ---
>
> W1: GCNeuron’s combinational logic is used only to describe concepts associated with individual neurons, rather than to construct explicit class-wise decision rules (e.g., $ p \land q \Rightarrow \text{Class 1} $).
>
> The resulting class-related concepts are quantified using numerical importance scores that measure how strongly a neuron distinguishes between classes. **However, these neuron importance scores are not translated into explicit class-wise logical rules; this differs from other rule-based explainers, including ours.** For this reason, we do not include it as a baseline.
>
> ---
>
> W2: We would like to clarify that **BAMultiShapes is indeed a synthetic benchmark with ground-truth logical rules**; see https://pytorch-geometric.readthedocs.io/en/2.5.0/generated/torch_geometric.datasets.BAMultiShapesDataset.html
> . Despite substantial noise in the underlying graph structure, our method can recover these rules accurately.
>
> To be more specific, the ground truth rule for Class 1 is $ (H \land W) \lor (H \land G) \lor (W \land G) \Rightarrow \text{Class 1} $, where $H$, $W$, $G$ denote the House, Wheel, and Grid motifs.
>
> As shown in our DNF rule explanations for Class 1 on BAMultiShapes (Appendix E.3), the extracted clause $(p_{52} \land p_{55})$ corresponds to $(H \land W)$, while $(p_{280} \land p_{113})$ corresponds to $(W \land G)$, when restricting rule depth to 5. The full rule can also be recovered when the depth is set to 10.
>
> **Thus, this shows the ability of rule learning rather than motif detection, which we have also used to validate our approach during development.** In contrast, SOTA baselines such as GLGExplainer and GraphTrail fail to recover comparable ground-truth rules on BAMultiShapes; see Fig. 5 in GraphTrail [1].
>
> ---
>
> W3 & W4: The grounded rules provide formally precise explanations that generalize beyond individual subgraphs and can explain structural patterns. Although they may appear complex, **informal feedback from chemistry domain experts suggests that they are human-understandable and useful in practice**. In particular, experts often reason in terms of rules and structural constraints, such as SMARTS-style substructure rules (https://www.daylight.com/dayhtml/doc/theory/theory.smarts.html) [2], especially when identifying chemically meaningful structural patterns (e.g., constraints on allowable atom types or bonding configurations). Such information cannot be revealed by individual subgraph explanations alone, which is why motif-based or generation-based explanations are less suitable, since they produce only subgraph-level explanations.
>
> **As we offer both representative subgraphs (for lightweight inspection) and fully grounded logical rules (for domain-level reasoning) as complementary explanations, we view this as a trade-off between simplicity and formal precision, intended to support different user needs.**
>
> ---
>
> W5 & W6: We acknowledge that our work may not be the first one to assess fidelity using subgraphs, as we believe this should be considered as standard practice. **Our novelty therefore lies in extending subgraph-based fidelity evaluation to rule-based explanations**, which prior methods did not adopt as an evaluation strategy, and consequently tend to overestimate explanation performance.
>
> **Another important novelty is our design of structure-aware predicates**, which jointly incorporate WL-hashing–based structural patterns (where message passing occurs) and the GNN’s embeddings. Preserving such structural information is essential for achieving reliable grounding, **as WL hashing ensures that non-isomorphic structures are never encoded by the same predicate.** Existing rule-based methods fall short in this respect; for example, GraphTrail relies on so-called “computation trees,” which have limited capacity to distinguish different graph structures, thereby preventing reliable grounding.
>
> Finally, as discussed in our initial rebuttal, we believe technical novelty should be evaluated based on concrete algorithmic and design advances rather than paradigm shifts alone, which are outside the main scope of this work.
>
>
> ---
>
> ### References
>
> [1] Armgaan et al., *GraphTrail: Translating GNN Predictions into Human-Interpretable Logical Rules,* NeurIPS 2024.
>
> [2] Daylight Chemical Information Systems, *SMARTS: A Language for Describing Molecular Patterns.*
> https://www.daylight.com/dayhtml/doc/theory/theory.smarts.html

---

> ### Author Response · Authors · 2025-11-28
> **A Follow-up on W3 & W4: Why Our Grounded Rule-Based Explanations Matter for Scientific Domains**
>
> ### A follow-up on the discussion of  W3 & W4: Why Our Grounded Rule-Based Explanations Matter for Scientific Domains
>
> *Response:* While we agree that different explanation paradigms are useful for understanding GNN behavior, they convey information at different levels of granularity and serve different end users. Although our grounded rule explanations may appear complex, informal feedback from practitioners in scientific domains suggests that this level of structural detail is both familiar and useful in practice.
>
> From a domain expert’s perspective (e.g., chemistry), our grounded rule-based explanations are preferred because **experts often reason in terms of rules and structural constraints, such as SMARTS-style substructure rules [1,2], rather than isolated instances.** Specifically, our grounded rule-based explanations describe general conditions that apply across many instances and can be checked against domain knowledge. In contrast, motif-based or generation-based explanations, such as TreeX and XGNN, produce only subgraph-level explanations—typically tied to individual instances rather than generalizable rule abstractions—which are less suitable in this setting.
>
> **Conceptually, the correspondence between SMARTS-style rules and our grounded rules can be summarized as follows (as an analogy rather than a syntactic equivalence):**
>
> | Aspect | SMARTS Rules | Our Grounded Rules |
> |------|-------------|------------------|
> | Structural constraints | Atom and bond constraints at pattern positions (e.g., `[O]`, `[C]-[O]`) | Orbit-wise node and edge constraints (e.g., `#O = 1`, `#(C,O) ≥ 2`) |
> | Local structural roles | Bond positions and local connectivity | Orbit indices encoding structural roles in the subgraph |
> | Variants | Logical OR over pattern variants | Multiple disjunctive rule clauses (DNF cases) |
> | Composition | Conjunctions of structural conditions | Conjunctions of predicates |
> | Generalization | Matches molecules not explicitly seen | Covers structural variants beyond observed motifs |
> | Semantics | Family of molecular structures | Family of grounded subgraphs (structural patterns) |
>
> In practice, this abstraction manifests concretely in our learned rules. For instance, in our DNF rules explaining BBBP (see Fig. 13), each clause (connected via OR operators) encodes a different oxygen-rich functional group or a combination of such groups associated with blood–brain barrier permeability. Within each clause, the rules further specify the internal structure of these functional groups, such as atom types, bonding patterns, and local connectivity, enabling a more precise and interpretable characterization of chemically meaningful structural patterns.
>
> This form of generalization in our grounded rules is consistent with how SMARTS rules and substructure queries are used in practice. For example, the trichloromethyl (CCl₃) group is a well-known structural alert associated with mutagenicity [3]. Our framework can capture trichloromethyl groups and related variants through grounded rules even when such substructures do not explicitly occur in the training dataset, because the learned rules encode structural constraints that generalize beyond individual observed motifs.
>
> In this context, the quality of grounded rule explanations is crucial. And since they are inferred using valid subgraphs (motifs) from the underlying dataset, using unreliable or ungrounded subgraphs can degrade the rule quality and mislead downstream analysis in practice. This observation motivates us to address an important gap in prior rule-based methods: achieving grounded explanations and eliminating chemically invalid substructures from explanations.
>
> Finally, while we did not include detailed domain-specific knowledge-discovery results in this submission, this choice reflects the general methodological focus of the ICLR audience. **One of our ongoing and future directions is to apply our framework to more challenging frontier biochemistry datasets to examine whether it can uncover previously unknown structure–activity relationships.** In this sense, we view our work as a complementary explanation approach that bridges methodological advances in GNN explainability with practical scientific insight.
>
> We hope these additional clarifications address your concerns regarding the positioning of our work and the motivation behind the rule-based, grounded explanation paradigm.
>
> ---
>
> ### References
>
> [1] D. Weininger, *SMILES, a Chemical Language and Information System*, Journal of Chemical Information and Computer Sciences, 1988.
>
> [2] Daylight Chemical Information Systems, *SMARTS: A Language for Describing Molecular Patterns*.
>     https://www.daylight.com/dayhtml/doc/theory/theory.smarts.html
>
> [3] E. Zeiger et al., *Salmonella Mutagenicity Tests: IV. Results from the Testing of 300 Chemicals*, Environmental and Molecular Mutagenesis, 1988.

---

### Official Review · Reviewer_kLwi · 2025-11-02

**Soundness:** 3
**Presentation:** 2
**Contribution:** 3
**Rating:** 8
**Confidence:** 3

**Summary:**

The paper addresses a real limitation in current global explainability methods for GNNs: they derive logical rules in a latent or concept space, and only afterwards associate these latent concepts with illustrative (sub)graphs; as a result, the generated example (sub)graphs often fail to correspond to real structures in the dataset (in molecular datasets, some are even chemically invalid).
The author propose a multi-step framework that learns logical rules whose predicates are directly grounded in subgraphs observable in the input data. The method is compared against two relatively recent baselines (GLGExplainer and GraphTrail), which the authors re-evaluate by substituting their latent concepts with the representative subgraphs provided by those methods. A new evaluation metric called data-grounded fidelity is introduced to assess how well the logical explanations reproduce the model’s behavior on real graphs rather than on latent representations.

Overall, the motivation is clear and the addressed problem appears real and relevant. However, the paper is occasionally difficult to follow, mainly because the pipeline uses several dense steps, sometimes similar (e.g. different decision trees for different purposes). The provided implementation also raises minor concerns: the released code explicitly skips the grounded part of the pipeline for all datasets except BBBP, Mutagenicity, and NCI1, and it appears not to run correctly for IMDB-BINARY.

**Strengths:**

The paper identifies a genuine limitation of current global explainability methods for GNNs: the lack of grounding of logical explanations in real graph instances. The proposed framework offers a systematic and sound solution. The proposed metric is reasonable. The presentation of the baseline results seems rigorous: tha appendix states that the authors of the original methods were consulted to verify the correctness of the reproduction.

**Weaknesses:**

The proposed procedure is quite convoluted, and the absence of ablation studies makes it difficult to understand which components of the pipeline are essential and which could be simplified. For similar reasons, the methododgical explanation is hard to follow: the paper requires several readings to fully grasp the role of each step (e.g., the multiple decision trees used for different purposes). The experimental comparison includes only two baselines.

**Questions:**

1. The released code explicitly skips the grounded part of the pipeline for all datasets excepts BBBP, Mutagenicity, and NCI1. Could the authors clarify whether this was intentional and whether the grounded evaluation can be extended to other datasets (e.g., IMDB-BINARY doesn’t seem to work)?
2. The procedure is difficult to follow. Have the authors considered simplifying the exposition, for example through a schematic overview?
3. An ablation study could help isolate which steps in the pipeline are most important. In addition, testing against more baselines would strengthen the empirical evidence.

---

> ### Author Response · Authors · 2025-11-14
> **Initial Rebuttal to the Review kLwi**
>
> We sincerely thank the reviewer for the positive and thoughtful assessment of our work, as well as for the constructive feedback. Below, we address each concern and question in detail.
>
>
> ---
>
>
> ### Q1: The released code explicitly skips the grounded part of the pipeline for all datasets except BBBP, Mutagenicity, and NCI1. Could the authors clarify whether this was intentional and whether the grounded evaluation can be extended to other datasets (e.g., IMDB-BINARY doesn’t seem to work)?
>
> *Response:*  Thank you for pointing out this issue. This behavior was not intentional. An oversight during the repository upload caused the script for handling several datasets to not be uploaded correctly. *We have now fixed the issue, and the grounded evaluation should run properly on additional datasets, including IMDB-BINARY.*
>
> We sincerely apologize for the confusion. In the final revision, we will also streamline the codebase and will consider releasing a dedicated Python package to ensure that the full pipeline is easier to use and more reliable.
>
>
>
> ---
>
> ### W1 & Q2: The proposed procedure feels convoluted, and the absence of ablation studies makes it difficult to see which components are essential. The methodological explanation is also hard to follow, for example, the multiple decision trees serving different purposes. Have the authors considered simplifying the exposition, such as by adding a schematic overview?
>
> *Response:* We appreciate the reviewer’s careful reading and fully understand that the exposition may feel dense due to the multi-step structure and the presence of several decision trees serving different purposes. We apologize for the difficulty this caused.
>
> In the final revision, we will improve the clarity of the methodological presentation by:
>
> 1. Adding a schematic overview that summarizes the pipeline at a high level.
>
> 2. Providing a clearer explanation of the role of each decision tree and component, highlighting their distinct purposes.
>
> 3. Streamlining the narrative to reduce redundancy and improve readability.
>
> We believe these changes will make the pipeline significantly easier to follow while preserving the rigor of the approach.
>
>
> ---
>
>
> ### Q3: An ablation study could help isolate which steps in the pipeline are most important. In addition, testing against more baselines would strengthen the empirical evidence.
>
> *Response:* Thank you for this valuable suggestion. We agree that an ablation study would clarify the contribution of each component in the pipeline, and we will include one in the final revision.
>
> Regarding baselines, our primary scope focused on grounding issues in existing *logic rule–based model-level* explainers. Through our literature review, we found that *only* two methods in this specific category are well established, namely GLGExplainer and GraphTrail. We appreciate the reviewer’s suggestion to broaden the comparison, and in the revision, we will additionally incorporate other forms of global explanation methods into the qualitative comparison to further strengthen the empirical evaluation.

---

> > ### Comment · Reviewer_kLwi · 2025-11-24
> > **Thanks!**
> >
> > Hi,
> >
> > Thanks for the clarifications. I will keep my positive score!

---

### Official Review · Reviewer_nxGX · 2025-11-03

**Soundness:** 3
**Presentation:** 3
**Contribution:** 3
**Rating:** 6
**Confidence:** 4

**Summary:**

This paper presents a framework to extract logic rules as explanation structures for GNN outputs. The method encodes node receptive fields using a WL kernel-based hash function and trains a decision tree to generate regulations that consistently classify the model's predictions. Conjunctive logic rules are extracted as grounding rules to link predicates with the input feature space. An experimental study verified the methods' efficiency, fidelity, and scalability.

**Strengths:**

S1. There is novelty in WL hashing-based structural encoding and decision tree for generating logic rules.
S2. The overall method is justified with a cost analysis.
S3. Results support the major claims.

**Weaknesses:**

W1. The link between the quality of the generated rules and their closeness to real-world evidence remains unevaluated.
W2. The time-cost analysis lacks a more rigorous elaboration. Important steps seem overly simplified or omitted.
W3. More baselines are needed, such as motif-based GNN explanation, which can directly generate explanations as subgraphs.

**Questions:**

D1. A main challenge is how to ensure the explanation structures are better grounded in genuine real-world evidence, while the process yields rules based on GNNs that still seem to focus on the models' faithfulness. Have any human experts or authorities assisted in evaluating the generated rules?  How will such a measure be quantified, if possible?

D2. The time cost omits several sources of overhead, such as training decision trees; a more complete analysis is not in place.

D3. Is the method model-specific? Meaning: if one changes to another test set or another model, does the method need to be restarted from scratch, even when the graph is not changed? How may the method respond to larger-scale analysis?

D4. There is a lack of in-depth analysis of how likely the rules are to be redundant or logically entailed by others—a missed opportunity for optimization?

D5. Other approaches, including motif-based GNN explanation, directly output graph patterns or subgraphs, which can be readily converted to conjunctive triple patterns or rules. Representative work needs to be compared with.

---

> ### Author Response · Authors · 2025-11-16
> **Initial Rebuttal to the Review nxGX (Question 1–2)**
>
> We sincerely thank the reviewer for the positive and thoughtful assessment of our work, as well as for the constructive feedback. Below, we address each concern and question in detail.
>
> ---
>
> ### W1 & Q1. The link between the quality of the generated rules and their closeness to real-world evidence remains unevaluated. Have any human experts or authorities assisted in evaluating the generated rules? How will such a measure be quantified, if possible?
>
> *Response:* Thank you for raising this important point. We agree that validating the closeness of the extracted rules to real-world evidence is essential. During method development, we examined this from two complementary angles:
>
> **(1) Validation on synthetic tasks with known ground truth.**
> We intentionally designed two controlled settings where the true decision rules are known:
>
> - **Case 1 (Fig. 2):** A simple GNN classification task where graphs 1–3 lack degree-2 nodes while graphs 4–5 contain them. Our method achieved 100% fidelity, and the extracted rule (Eq. 11) exactly matches the ground-truth decision boundary.
>
> - **Case 2 (BAMultiShapes; Tab. 1):**  A more structured controlled benchmark with ground-truth rules over House (H), Wheel (W), and Grid (G) motifs. Our approach correctly recovers these motif rules (Appendix E.3).
>
> **(2) Expert validation on domain-specific datasets.**
> For molecular datasets (BBBP and Mutagenicity), we consulted domain experts to verify whether the extracted rules align with established chemical knowledge. The extracted patterns indeed match well-known SARs (Structure–Activity Relationships):
>
> - In **BBBP**, our rules identifying oxygen-rich functional groups as predictive of non-permeability are consistent with established chemical principles: such groups increase hydrophilicity, decrease lipophilicity, and are more likely to be recognized by efflux transporters.
>
> - In **Mutagenicity**, our method recovers the well-known **nitro group (NO₂)** on aromatic rings, widely recognized for its association with DNA damage and mutagenesis, and also identifies other alert substructures such as the **trichloromethyl group**.
>
> These findings indicate that our rules are not only model-faithful but also chemically accurate. *In contrast, existing baseline methods frequently struggle to recover chemically meaningful motifs and rarely align with known SARs.*
>
> **The reason we did not include a large-scale human evaluation study is that such scientific validation typically does not rely on subjective human opinions, as knowledge discovery in this setting is objective and evidence-driven rather than perceptual.** Additional considerations are discussed in Appendix E.2. Accordingly, in addition to the above qualitative validation by domain experts, we also use objective, quantitative metrics—*coverage, stability, and validity*—to evaluate the quality of our generated explanations.
>
> At the same time, we view systematic, expert-driven evaluation as an important direction for future work. In particular, we plan to further investigate the **knowledge-discovery capabilities** of our framework in molecular domains through close collaboration with domain experts, who can rigorously assess the scientific relevance of the discovered rules and help refine our framework for practical scientific discovery.
>
> We will further clarify these points in the final revision.
>
> ---
>
> ### W2 & Q2. The time cost omits several sources of overhead, such as training decision trees; a more complete analysis is not in place.
>
> *Response:* We acknowledge the reviewer’s concern. Due to page constraints, our runtime discussion focused on the dominant components of the method, so several secondary overheads were not reported individually. Below, we clarify the cost of one such component, training the grounding decision trees.
>
> For each predicate $p$, we construct a dataset of representative subgraphs and train a small, shallow decision tree on low-dimensional structural features. Let $N_p$ denote the number of training examples and $d_p$ the feature dimension. Standard CART-style training has complexity
>
> $$O(d_p N_p \log N_p).$$
>
> In our setting:
> - $d_p$ is a small constant (simple predicate-level statistics),
> - $N_p \le |{\mathcal V}|$ because each example corresponds to a grounded subgraph, and
> - the total number of predicates is at most $O(|{\mathcal V}|)$.
>
> Summing the cost across all predicates gives
>
> $$\sum_p O(d_p N_p \log N_p) \le O(|{\mathcal V}|^2 \log |{\mathcal V}|).$$
>
> This is consistent with the $O(|{\mathcal V}|^2)$ grounding term reported in the paper; the extra $\log |{\mathcal V}|$ factor is mild, and because the trees are intentionally shallow and $d_p$ is constant, this overhead remains small relative to subgraph extraction and predicate construction.
>
> In the revision, we will include a clearer and more comprehensive analysis of the pipeline’s computational cost.

---

> ### Author Response · Authors · 2025-11-16
> **Initial Rebuttal to the Review nxGX (Question 3–5)**
>
> ### Q3. Is the method model-specific? Meaning: if one changes to another test set or another model, does the method need to be restarted from scratch, even when the graph is not changed? How may the method respond to larger-scale analysis?
>
> *Response:*  **As a model explainer, the method is designed to align with the underlying predictive model.** Because it uses the GNN’s activation patterns to construct predicates, it is indeed model-specific: if the model changes, the predicates and resulting rules should be recomputed to reflect the new decision boundaries.
>
> That said, the procedure does not need to be rerun for every scenario. If the dataset remains the same and only the test split changes, the extracted predicates and rules can be reused. Even across models trained on different data splits, the extracted rules remain highly similar (high stability in Tab. 4), *indicating that they remain effective across runs*, as well-trained GNNs typically converge to comparable decision boundaries.
>
> Regarding larger-scale analysis, our runtime study shows that all components scale polynomially, and empirical results on large benchmarks (Tab. 2) indicate that our method remains tractable while other baselines frequently time out. Thus, it is expected to run well under larger-scale settings.
>
> ---
>
> ### Q4. There is a lack of in-depth analysis of how likely the rules are to be redundant or logically entailed by others—a missed opportunity for optimization?
>
> *Response:*  We thank the reviewer for this insightful question, which *we also considered during method development*.
>
> **From a purely logical perspective, the extracted rules are unlikely to be redundant or logically entailed by one another.** Each predicate is defined via a WL-hash over local neighborhoods, and distinct WL-hashes correspond to distinct structural patterns; therefore, predicates do not logically subsume one another by construction. At the rule level, the decision tree that determines the logical structure further mitigates redundancy: it selects a minimal subset of predicates necessary for class separation, and different branches correspond to mutually exclusive conditions, preventing overlap in the resulting rules.
>
> At a semantic level, it is *theoretically possible* for different predicates to capture related or partially overlapping substructures. However, in practice, our sanity checks during development did not reveal meaningful redundancy among predicates extracted from well-trained models. A plausible explanation is that well-trained GNNs tend to develop sparse and efficient internal representations—focusing on a small set of informative patterns rather than encoding multiple redundant variants of the same structure. As a result, the extracted rules naturally remain concise and non-redundant.
>
> In the revision, we will incorporate a brief discussion of this topic.
>
>
> ---
>
>
> ### W3 & Q5. Other approaches, including motif-based GNN explanation, directly output graph patterns or subgraphs, which can be readily converted to conjunctive triple patterns or rules. Representative work needs to be compared with.
>
> *Response:*  We thank the reviewer for this helpful suggestion. *During development, we did consider using motif-based GNN explanation methods as baselines for quantitative comparison.* However, we found that existing motif-based approaches **cannot** be readily transformed into **class-wise logical rules with a clear (non-overlapping) decision boundary**.
>
> More specifically, motif-based methods such as XGNN (Yuan et al., 2020), D4Explainer (Chen et al., 2024), and GNNInterpreter (Wang & Shen, 2022) generate representative motifs or subgraphs for a target class, but they do not elucidate the model’s decision boundary. When attempting to convert these motifs into class-wise logical rules, we observed that a single graph can often match motifs associated with **multiple** classes, making it impossible to derive non-overlapping logical rules as in our work, or to use such rules to obtain definitive class predictions for arbitrary graph inputs. As a consequence, these approaches cannot support data-grounded fidelity evaluation, *unlike logic rule–based methods such as GLGExplainer, GraphTrail, and our work.*
>
>
> For these reasons, motif-based methods are not directly comparable to ours in terms of quantitative fidelity. Nevertheless, we agree that they represent an important category of explanation techniques, and in the final revision, we will include a qualitative comparison with representative motif-based approaches to more clearly situate our method within the broader GNN explainability landscape.

---

### Official Review · Reviewer_AscN · 2025-11-03

**Soundness:** 4
**Presentation:** 4
**Contribution:** 4
**Rating:** 8
**Confidence:** 3

**Summary:**

The paper proposes a novel approach to post-hoc explanations of graph classification, based on concepts. Differently from previous methods, however, it grounds the explanations into actual patterns in the data by means of Weisfeiler–Lehman (WL) graph hashing.

**Strengths:**

- The paper is well written and presented.
- The methodology is not particularly complex, but it allows creating a very effective method
- Section 3.4 reports a very nice Analysis subsection, theoretically describing the computational complexity of the method and its applicability to different gnn architectures
- The experiments show a clear advantage of the proposed method against a couple of state-of the-art baselines

**Weaknesses:**

General weaknesses:
- The paper does not have a limitation paragraph, which is now considered almost mandatory in top-level conferences
- The scope of the proposed method is focused to explanations regarding graph classification only (similarly to the baselines). It would be very interesting if the authors could show or even just describe whether the proposed method could be applied to node classification, or if any methodological assumptions fail to hold in that case.
- Also in several parts there are mentions of graph tasks, suggesting that different types of tasks have been considered, which does not seem the case: only multiple instances of the graph classification task have been tested. I would recommend rephrasing as it is currently misleading.
- One of the main result of the paper is that previous baselines provide explanations that are not grounded. However, in the main paper (I saw them in the appendix) there are no mentions regarding how the baselines have been reproduced. Without at least a footnote saying how the explanations for the baselines have been extracted, Figure 1 results too strong and may rise critiques.


Specific weaknesses:
- The sentence "As a result, LOGICXGNN not only generates a rich set of representative subgraphs but also learns generalizable grounding rules for each predicate, addressing unreliable grounding in existing methods." is not very clear I would suggest rephrasing.
- $p_j$ and $P$ are not properly defined

**Questions:**

My main question is regarding the applicability of the proposed method to node classification task:
- is it feasible to consider the same framework also in this case?

My guess is that the hashing could be re-used similarly but possibly also the decision trees to select the patterns over the embeddings and the one over the activation matrix.

---

> ### Author Response · Authors · 2025-11-13
> **Initial Rebuttal to the Review AscN**
>
> We sincerely thank the reviewer for the positive assessment of our work and for the constructive, insightful feedback. Below, we address each concern and question in detail.
>
> ---
>
> ### W1: The paper does not have a limitation paragraph, which is now considered almost mandatory in top-level conferences.
>
> *Response:*  Thank you for pointing this out. We will include a dedicated limitations section in the final version. Specifically, we plan to discuss two main aspects:
> 1. Although our extracted logical rules achieve high fidelity, they do not yet *perfectly* mirror the full decision-making process of GNNs. We aim to further improve this alignment in future work.
> 2. The presentation of explanations to end users can be further enhanced. In particular, for domain-specific datasets such as molecular graphs, we plan to incorporate *domain knowledge* to make the final explanations more intuitive and semantically meaningful for practitioners.
>
> ---
>
> ### W2 & Q1: The scope of the proposed method is focused on graph classification only (similarly to the baselines). It would be interesting if the authors could show or describe whether the proposed method could be applied to node classification, or if any methodological assumptions fail to hold in that case.
>
> *Response:*  We appreciate the reviewer’s insightful question regarding the potential extension of our framework to node classification. Indeed, our approach *naturally supports node-level tasks* with only minor modifications. Specifically, the first and third stages (*predicate formulation* and *predicate grounding*) remain directly applicable, as they operate at the node level rather than over entire graphs. The only step specific to graph-level tasks is the activation-matrix construction (Step 2), which is used exclusively for learning graph-level rules.
>
> Recall that the predicate function  $f(v) = (Pattern_{struct}(v), Pattern_{emb}(v))$ encodes class-informative signals at the node level:  $Pattern_{emb}(v)$ is derived from class-separating embedding dimensions identified by the decision tree, ensuring alignment with the GNN’s class distinctions. *Therefore, for node classification, class-wise rules can be obtained simply by combining the predicates associated with each class through logical OR conditions*. As a result, the framework can be adapted to node classification without violating any methodological assumptions.
>
>
> In earlier versions of the paper, we included preliminary results on node classification but later removed them due to page constraints and the lack of suitable baseline comparisons, as existing rule-based explanation methods generally do not support node-level tasks. We thank the reviewer for this suggestion and will include these results in the final revision.
>
>
> ---
>
> ### W3: The paper mentions multiple “graph tasks,” which may be misleading since only graph classification has been evaluated.
>
> *Response:* We thank the reviewer for the careful reading and valuable feedback. You are correct that the current phrasing may appear misleading. We originally included node-classification results in earlier drafts, but after removing them due to space constraints, some of the associated phrasing was inadvertently left unchanged. We acknowledge this oversight. In the final revision, we plan to *reintroduce node classification results* and *refine the text* to more accurately reflect the evaluated task types and avoid potential confusion.
>
> ---
>
> ### W4: One of the main results of the paper is that previous baselines provide explanations that are not grounded. However, in the main paper (I saw them in the appendix) there are no mentions regarding how the baselines have been reproduced. Without at least a footnote saying how the explanations for the baselines have been extracted, Figure 1 results too strong and may rise critiques.
>
> *Response:*  We thank the reviewer for this helpful suggestion. All baseline experiments were reproduced using the authors’ official implementations, and we consulted the original authors to ensure a fair and faithful diagnosis of the grounding issues observed in their methods. The full reproduction details are provided in the appendix due to their length. We agree that adding a brief footnote in the main paper regarding this process will improve clarity and address potential concerns, and we will include it in the revision.
>
> ---
>
> ### Other weaknesses:
>
> 1. The sentence "As a result, LOGICXGNN not only generates a rich set of representative subgraphs but also learns generalizable grounding rules for each predicate, addressing unreliable grounding in existing methods." is not very clear and should be rephrased.
> 2. $p_j$ and $P$ are not properly defined.
>
> *Response:*  We thank the reviewer for the helpful suggestions. We will rephrase the sentence for clarity and properly define both $p_j$ and $P$ in the revision.

---

### Author Response · Authors · 2025-11-30
**Summary for Area Chair**

Thank you for considering our submission. Below is a concise summary to aid the meta-review.

**Contribution.**
LogicXGNN is a post-hoc grounded rule-based global explainer for GNNs. Our key contributions are:

1. **Motivated by a genuine and important gap.**
We identify that existing rule-based explainers often lack proper grounding; they tend to overestimate fidelity and can produce unreliable explanations, such as chemically invalid substructures that do not exist in molecular datasets. This can mislead end users, particularly in scientific domains.


2. **Novel design to achieve effective grounding.**
We introduce a novel structure-aware predicate design based on WL hashing, ensuring that both subgraph explanations and logical rules correspond to valid structures observed in the data, thereby achieving effective grounding.


3. **Strong empirical performance.**
Across benchmarks, LogicXGNN improves fidelity by over **20% on average** compared to SOTA rule-based explainers while being **10–100× faster** and remaining scalable. Both quantitative and qualitative analyses demonstrate that our method produces higher-quality explanations than baselines.


**Review context.**
Three reviewers recommend acceptance (**scores: 8, 8, 6**),  highlighting strong motivation, novelty in WL hashing-based structural grounding, and solid empirical performance. One reviewer, kSni (**score: 4**), also acknowledges our motivation, methodological effectiveness, and completeness, and strong empirical results, but raises concerns primarily regarding baseline selection (W1, W5), the lack of validation on synthetic benchmarks (W2), the choice of explanation paradigm (W3, W4), and perceived limitations in technical novelty (W6).

**Response to concerns.**
In the rebuttal and follow-up:
- **Baseline selection (W1, W5):**
We clarified that the two works suggested by Reviewer kSni adopt different explanation paradigms and cannot be readily converted into class-wise logical rule explanations used in our framework; therefore, they are not suitable baselines. We also clarified our baseline choices to Reviewers nxGX and kLwi—Reviewer kLwi accepted this clarification, and Reviewer nxGX raised no further questions.

- **The lack of validation on synthetic benchmarks (W2):**
This appears to be a misunderstanding. During method development, we validated our rule explanations against known ground-truth rules on synthetic benchmarks. Results on the widely used BAMultiShapes benchmark are included in the paper.

- **The choice of explanation paradigm (W3, W4):**
We clarified that our method is designed to support two complementary explanation modes: (1) rules over representative subgraphs that match the format of prior methods, and (2) fully grounded logical rules that provide formal precision when desired. In particular, (2) aligns with *SMARTS/SAR-style* reasoning used by chemistry domain experts (details are provided in comments). Informal feedback from such practitioners supports the usefulness of this explanation form, and we are actively refining the presentation to better match their needs. Reviewer kSni’s main concern is that (2) may be too complex; however, we view this concern largely as reflecting differing domain preferences rather than a methodological deficiency.

- **Technical novelty (W6):**
We clarified our technical novelty in detail and explained how our design achieves effective grounding while remaining efficient and scalable. Reviewer kSni’s comments on novelty appear to be made at a meta level and are inconsistent with the assessments of the other reviewers, all of whom found our work to be novel and technically meaningful.

**Motivation and future work.**
This work is motivated by our interest in applying advances in XAI to scientific discovery, where we found that many existing GNN explainers are not well-suited for practical use. In particular, while rule-based paradigms are preferable in scientific settings because they provide formally precise and semantically clear logical explanations, existing methods often lack faithfulness and may produce misleading or incorrect explanations.

*One of our ongoing and future directions is to apply our framework to more challenging frontier biochemistry datasets to examine whether it can uncover previously unknown structure–activity relationships.* In this sense, we view this work as our first and important step in this journey. We also did not include preliminary domain-specific knowledge-discovery results in this submission; this choice reflects the general methodological focus of the ICLR audience.

In short, we view our work as a complementary explanation approach that bridges methodological advances in GNN explainability with practical scientific insight. *Our code is publicly available, and we hope it contributes to the XAI community and supports future knowledge discovery.*

We hope this summary helps contextualize the reviews and our responses.

---

### Meta-Review · Area_Chair_gjrr · 2025-12-08

**Summary:**

Three reviewers (AscN, nxGX, kLwi) gave scores of 8, 6, and 8, respectively, praising the motivation, novelty of grounding via WL hashing, and empirical strength. Reviewer kSni gave a 4, raising concerns about baseline selection, synthetic validation, explanation complexity, and technical novelty. The authors responded thoroughly in rebuttal, addressing each point with clarifications, synthetic results (including BAMultiShapes with ground-truth rules), and domain-expert validation.
Therefore, I think the authors have addressed the core concerns raised by the reviewers, and the work exhibits a clear problem motivation, solid technical innovation, and reliable empirical support—making it suitable for acceptance at ICLR.

**Reviewer Scores:**

AscN:8.
nxGX:6.
kLwi:8.
kSni:4.

---

### Decision · Program_Chairs · 2026-01-26

Accept (Poster)